# ONLINE PSEUDO-ZEROTH-ORDER TRAINING OF NEUROMORPHIC SPIKING NEURAL NETWORKS

**Mingqing Xiao**[1,2]**, Qingyan Meng**[3]**, Zongpeng Zhang**[4]**, Di He**[2,5]**, Dongsheng Li**[1]**,
Zhouchen Lin**[2,5*]

[1]Microsoft Research Asia
[2]State Key Lab of General AI, School of Intelligence Science and Technology, Peking University
[3]Peng Cheng Laboratory
[4]Department of Biostatistics, School of Public Health, Peking University
[5]Institute for Artificial Intelligence, Peking University
{mingqingxiao, dongsli}@microsoft.com, mengqy@pcl.ac.cn
zongpeng.zhang98@gmail.com, {dihe, zlin}@pku.edu.cn

## ABSTRACT

Brain-inspired neuromorphic computing with spiking neural networks (SNNs) is a promising energy-efficient computational approach. However, successfully training deep SNNs in a more biologically plausible and neuromorphic-hardware-friendly way is still challenging. Most recent methods leverage spatial and temporal backpropagation (BP), not adhering to neuromorphic properties. Despite the efforts of some online training methods, tackling spatial credit assignments by alternatives with competitive performance as spatial BP remains a significant problem. In this work, we propose a novel method, online pseudo-zeroth-order (OPZO) training. Our method only requires a single forward propagation with noise injection and direct top-down signals for spatial credit assignment, avoiding spatial BP's problem of symmetric weights and separate phases for layer-by-layer forward-backward propagation. OPZO solves the large variance problem of zeroth-order methods by the pseudo-zeroth-order formulation and momentum feedback connections, while having more guarantees than random feedback. Combining online training, OPZO can pave paths to on-chip SNN training. Experiments on neuromorphic and static datasets with both fully connected and convolutional networks demonstrate the effectiveness of OPZO with competitive performance compared with spatial BP, as well as estimated low training costs.

## 1 INTRODUCTION

Neuromorphic computing with biologically inspired spiking neural networks (SNNs) is an energy-efficient computational framework with increasing attention recently (Roy et al., 2019; Schuman et al., 2022). Imitating biological neurons to transmit spike trains for sparse event-driven computation as well as parallel in-memory computation, efficient neuromorphic hardware is developed, supporting SNNs with low energy consumption (Davies et al., 2018; Pei et al., 2019; Woźniak et al., 2020; Rao et al., 2022; Davies, 2021).

Nevertheless, supervised training of SNNs is challenging considering neuromorphic properties. While popular surrogate gradient methods can deal with the non-differentiable problem of discrete spikes (Shrestha & Orchard, 2018; Wu et al., 2018; Neftci et al., 2019), they rely on backpropagation (BP) through time and across layers for temporal and spatial credit assignment, which is biologically problematic and would be inefficient on hardware.

Particularly, spatial BP suffers from problems of weight transport and separate forward-backward stages with update locking (Crick, 1989; Frenkel et al., 2021), and temporal BP is further infeasible for spiking neurons with the online property (Bellec et al., 2020). Considering learning in biological systems with unidirectional local synapses, maintaining reciprocal forward-backward connections

---

*corresponding author

with symmetric weights and separate phases of signal propagation is often viewed as biologically problematic (Nøkland, 2016), and also poses challenges for efficient on-chip training of SNNs. Methods with only forward passes, or with direct top-down feedback signals acting as modulation in biological three-factor rules (Frémaux & Gerstner, 2016; Roelfsema & Holtmaat, 2018), are more efficient and plausible, e.g., on neuromorphic hardware (Davies, 2021).

Some previous works explore alternatives for temporal and spatial credit assignment. To deal with temporal BP, online training methods are developed for SNNs (Bellec et al., 2020; Xiao et al., 2022). With tracked eligibility traces, they decouple temporal dependency and support forward-in-time learning. However, alternatives to spatial BP still require deeper investigations. Most existing works mainly rely on random feedback (Nøkland, 2016; Bellec et al., 2020), with limited guarantees and poorer performance than spatial BP. Some works explore forward gradients (Silver et al., 2022; Baydin et al., 2022), but they require an additional stage of heterogeneous signal propagation and perform poorly due to the large variance. Recently, Malladi et al. (2023) show that zeroth-order (ZO) optimization with simultaneous perturbation stochastic approximation (SPSA) can effectively fine-tune pre-trained large language models, but it requires specially designed settings, not suitable for general neural network training due to the large variance. On the other hand, local learning has been studied, e.g., with local readout layers (Kaiser et al., 2020) or forward-forward self-supervised learning (Hinton, 2022; Ororbia, 2023). It is complementary to global learning and can improve some methods (Ren et al., 2023). As a crucial component of machine learning, efficient global learning alternatives with competitive performance remain an important problem.

In this work, we propose a novel online pseudo-zeroth-order (OPZO) training method with only a single forward propagation and direct top-down feedback for global learning. We first propose a pseudo-zeroth-order formulation for neural network training, which decouples the model and loss function and maintains the zeroth-order formulation for neural networks while leveraging the available first-order property of the loss function for more informative feedback error signals. Then we propose momentum feedback connections to directly propagate feedback signals to hidden layers. The connections are updated based on the one-point zeroth-order estimation of the expectation of the Jacobian, with which the large variance of zeroth-order methods can be solved, and more guarantees are maintained compared with random feedback. OPZO only requires a noise injection in the common forward propagation, flexibly applicable to black-box or non-differentiable models. Built upon online training, OPZO enables training in a similar form as the three-factor Hebbian learning based on direct top-down modulations, paving paths to on-chip training of SNNs. Our contributions include:

1. We propose a pseudo-zeroth-order formulation that decouples the model and loss function for neural network training, which enables more informative feedback signals while keeping the zeroth-order formulation of the (black-box) model.

2. We propose the OPZO training method with a single forward propagation and momentum feedback connections, solving the large variance of zeroth-order methods and keeping low costs. Built on online training, OPZO provides a more biologically plausible method friendly for potential on-chip training of SNNs.

3. We conduct extensive experiments on neuromorphic and static datasets with fully connected and convolutional networks, as well as on ImageNet with larger networks fine-tuned under noise. Results show the effectiveness of OPZO in reaching competitive performance compared with spatial BP and its robustness under different noise injections. OPZO is also estimated to have lower computational costs than BP on potential neuromorphic hardware.

## 2 RELATED WORK

**SNN Training Methods** A mainstream method is spatio-temporal BP combined with surrogate gradient (SG) (Shrestha & Orchard, 2018; Wu et al., 2018; Neftci et al., 2019), with many efforts on architecture or objective design (Yao et al., 2024; Xiao et al., 2024b; Lv et al., 2024; Deng et al., 2023; Guo et al., 2024; Xing et al., 2025). Another direction is to derive closed-form transformations or implicit equilibriums between encodings of spike trains (weighted firing rates or the first time to spike), and convert artificial neural networks (ANNs) to SNNs (Rueckauer et al., 2017; Deng & Gu, 2021; Stöckl & Maass, 2021; Meng et al., 2022b) or directly train SNNs with gradients from the transformations (Zhou et al., 2021; Wu et al., 2021; Meng et al., 2022a) or equilibriums (Xiao et al., 2021; Martin et al., 2021; Xiao et al., 2023). To tackle the problem of temporal BP, some online

training methods are proposed (Bellec et al., 2020; Xiao et al., 2022; Bohnstingl et al., 2022; Meng et al., 2023; Yin et al., 2023) for forward-in-time learning, but many of them still require spatial BP. Considering alternatives to spatial BP, Neftci et al. (2017); Lee et al. (2020); Bellec et al. (2020) apply random feedback, Kaiser et al. (2020) propose online local learning, and Yang et al. (2022) propose local tandem learning with ANN teachers. Different from them, we propose a new global learning method with similar performance as spatial BP. Li et al. (2021) and Mukhoty et al. (2023) study zeroth-order properties for each parameter or neuron to adjust surrogate functions or leverage a local zeroth-order estimator for the Heaviside step function, lying in the spatio-temporal BP framework. Differently, in this work, zeroth-order training refers to simultaneous perturbation for global network training without BP.

**Alternatives to Spatial Backpropagation** For more biologically plausible global learning, alternatives to spatial BP are proposed. Target propagation (Lee et al., 2015), feedback alignment (FA) (Lillicrap et al., 2016), and sign symmetry (Liao et al., 2016; Xiao et al., 2018) avoid the weight symmetry problem by propagating targets or using random / only sign-shared backward weights, and Akrout et al. (2019) improves FA by learning it to be symmetric with forward weights. They, however, still need an additional stage of sequential layer-by-layer backward propagation. Direct feedback alignment (DFA) (Nøkland, 2016; Launay et al., 2020) improves FA to directly propagate errors from the last layer to hidden ones. However, random feedbacks have limited guarantees and perform much worse than BP. Some recent works study forward gradients (Silver et al., 2022; Baydin et al., 2022; Ren et al., 2023; Xiao et al., 2024a; Bacho & Chu, 2024), but they require an additional heterogeneous signal propagation stage, suffering from biological plausibility issues and larger costs. There are also methods focusing on energy functions (Scellier & Bengio, 2017) or lifted proximal formulation (Li et al., 2020). Besides global supervision, some works turn to local learning, using local readout layers (Kaiser et al., 2020), forward-forward contrastive learning (Hinton, 2022), or Hebbian learning (Journé et al., 2023). This work mainly focuses on global learning and can be combined with local learning.

**Zeroth-Order Optimization** ZO optimization has been widely studied in machine learning, such as for black-box optimization (Grill et al., 2015), adversarial attacks (Chen et al., 2017), reinforcement learning (Salimans et al., 2017), etc., at relatively small scales, but its application to direct neural network training is limited due to the variance caused by a large number of parameters. Recently, Yue et al. (2023) theoretically shows that the complexity of ZO optimization can exhibit weak dependencies on dimensionality considering the effective dimension, and Malladi et al. (2023) proposes zeroth-order SPSA for memory-efficient fine-tuning pre-trained large language models with a similar theoretical basis. However, it depends on specially designed settings (e.g., fine-tuning under the prompt setting (Malladi et al., 2023; Gautam et al., 2024)) which are not applicable to general neural network training, and requires two forward passes. Jiang et al. (2024) proposes a likelihood ratio method to train neural networks, but it requires multiple forward propagation proportional to the layer number in practice. Chen et al. (2024) considers the finite difference for each parameter rather than simultaneous perturbation and proposes pruning methods for improvement, limited in computational complexity. Differently, we propose a method for neural network training from scratch with only one forward pass for low costs and comparable performance to spatial BP.

# 3 PRELIMINARIES

## 3.1 SPIKING NEURAL NETWORKS

Imitating biological neurons, each spiking neuron keeps a membrane potential $u$, integrates input spike trains, and generates a spike for information transmission once $u$ exceeds a threshold. $u$ is reset to the resting potential after a spike. We consider the commonly used leaky integrate and fire (LIF) model with the dynamics of the membrane potential as: $\tau_m \frac{du}{dt} = -(u - u_{rest}) + R \cdot I(t)$, for $u < V_{th}$, with input current $I$, threshold $V_{th}$, resistance $R$, and time constant $\tau_m$. When $u$ reaches $V_{th}$ at time $t^f$, the neuron generates a spike and resets $u$ to zero. The output spike train is $s(t) = \sum_{t^f} \delta(t - t^f)$.

SNNs consist of connected spiking neurons. We consider the simple current model $I_i(t) = \sum_j w_{ij} s_j(t) + b_i$, where $i, j$ represent the neuron index, $w_{ij}$ is the weight and $b_i$ is a bias. The

discrete computational form is:

$$\begin{cases} u_i\,[t+1] = \lambda(u_i[t] - V_{th}s_i[t]) + \sum_j w_{ij}s_j[t] + b_i, \\ s_i[t+1] = H(u_i\,[t+1] - V_{th}). \end{cases} \quad (1)$$

Here $H(x)$ is the Heaviside step function, $s_i[t]$ is the spike signal at discrete time step $t$, and $\lambda < 1$ is a leaky term (taken as $1 - \frac{1}{\tau_m}$). For multi-layer networks, we use $\mathbf{s}^{l+1}[t]$ to represent the $(l+1)$-th layer's response after receiving signals $\mathbf{s}^l[t]$ from the $l$-th layer, i.e., the expression is $\mathbf{u}^{l+1}[t+1] = \lambda(\mathbf{u}^{l+1}[t] - V_{th}\mathbf{s}^{l+1}[t]) + \mathbf{W}^l\mathbf{s}^l[t+1] + \mathbf{b}^l$.

**Online Training of SNNs**  We build the proposed OPZO on online training methods for forward-in-time learning. Here online training refers to online through the time dimension of SNNs (Bellec et al., 2020; Xiao et al., 2022), as opposed to backpropagation through time. We consider OTTT (Xiao et al., 2022) to online calculate gradients at each time by the tracked presynaptic trace $\hat{\mathbf{a}}^l[t] = \sum_{\tau \leq t} \lambda^{t-\tau}\mathbf{s}^l[\tau]$ and instantaneous gradient $\mathbf{g}_{\mathbf{u}^{l+1}}[t] = \left( \frac{\partial \mathcal{L}[t]}{\partial \mathbf{s}^N[t]} \prod_{i=0}^{N-l-2} \frac{\partial \mathbf{s}^{N-i}[t]}{\partial \mathbf{s}^{N-i-1}[t]} \frac{\partial \mathbf{s}^{l+1}[t]}{\partial \mathbf{u}^{l+1}[t]} \right)^\top$ as $\nabla_{\mathbf{W}^l}\mathcal{L}[t] = \mathbf{g}_{\mathbf{u}^{l+1}}[t]\hat{\mathbf{a}}^l[t]^\top$. In OTTT, the instantaneous gradient requires layer-by-layer spatial BP with surrogate derivatives for $\frac{\partial \mathbf{s}^l[t]}{\partial \mathbf{u}^l[t]}$. The proposed OPZO, on the other hand, leverages only one forward propagation across layers and direct feedback to estimate $\mathbf{g}_{\mathbf{u}^{l+1}}[t]$ without spatial BP combining surrogate gradients.

## 3.2 Zeroth-Order Optimization

Zeroth-order optimization is a gradient-free method using only function values. A classical ZO gradient estimator is SPSA (Spall, 1992), which estimates the gradient of parameters $\boldsymbol{\theta}$ for $\mathcal{L}(\boldsymbol{\theta})$ on a random direction $\mathbf{z}$ as:

$$\nabla^{ZO}\mathcal{L}(\boldsymbol{\theta}) = \frac{\mathcal{L}(\boldsymbol{\theta} + \alpha\mathbf{z}) - \mathcal{L}(\boldsymbol{\theta} - \alpha\mathbf{z})}{2\alpha}\mathbf{z} \approx \mathbf{z}\mathbf{z}^\top\nabla\mathcal{L}(\boldsymbol{\theta}), \quad (2)$$

where $\mathbf{z}$ is a multivariate variable with zero mean and unit variance, e.g., following the multivariate Gaussian distribution, and $\alpha$ is a perturbation scale. Alternatively, we can use the one-sided formulation for this directional gradient $\frac{\mathcal{L}(\boldsymbol{\theta} + \alpha\mathbf{z}) - \mathcal{L}(\boldsymbol{\theta})}{\alpha}\mathbf{z}$. These two-point estimations are unbiased estimator of $\nabla\mathcal{L}(\boldsymbol{\theta})$ in the limit $\alpha \to 0$ under the common assumptions of $L$-smoothness of $\mathcal{L}(\boldsymbol{\theta})$ and i.i.d. components of $\mathbf{z}$ with zero mean and unit variance (Nesterov & Spokoiny, 2017; Duchi et al., 2015).

Considering biological plausibility and efficiency, estimation with a single forward pass is more appealing. Actually, we can leverage the single-point zeroth-order estimation ($ZO_{sp}$):

$$\nabla^{ZO_{sp}}\mathcal{L}(\boldsymbol{\theta}) = \frac{\mathcal{L}(\boldsymbol{\theta} + \alpha\mathbf{z})}{\alpha}\mathbf{z}. \quad (3)$$

For non-zero $\alpha$ in practice, it has the same expectation as the two-point method. Additionally, when $z$ is uniformly sampled from the unit sphere, the single-point estimation is an unbiased estimator of the smooth version of $\mathcal{L}$: $\mathcal{L}_\alpha(x) = \mathbb{E}_{z \in \mathbb{S}^n}[\mathcal{L}(\theta + \alpha\mathbf{z})]$, which does not require $\mathcal{L}$ to be differentiable (Flaxman et al., 2005).

The above formulation only requires a noise injection in the forward propagation, and the gradients can be estimated with a top-down feedback signal, as shown in Fig. 1(d). This is also similar to REINFORCE (Williams, 1992) and Evolution Strategies (Salimans et al., 2017) in reinforcement learning, and is considered to be biologically plausible (Fiete & Seung, 2006). It is believed that the brain is likely to employ perturbation methods for some kinds of learning (Lillicrap et al., 2020).

However, zeroth-order methods usually suffer from a large variance, since two-point methods only estimate gradients in a random direction and the one-point formulation has even larger variances. Therefore they hardly work for general neural network training. In the following, we propose our momentum-based pseudo-zeroth-order method to solve the problem, also only based on one forward propagation with noise injection and top-down feedback signals.

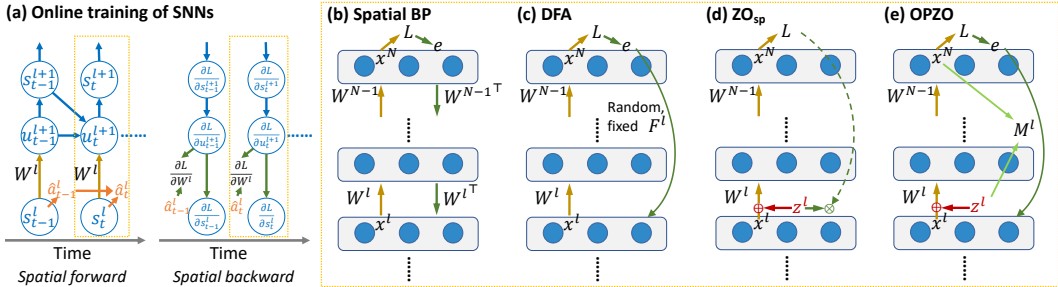

Figure 1: Illustration of different training methods. (a) Online training of SNNs with tracked traces for temporal credit assignment. (b-e) Different spatial credit assignment methods. (b) Spatial BP propagates errors layer-by-layer with symmetric weights. (c) DFA directly propagates error signals from the top layer to the middle ones with fixed random connections. (d) Single-point zeroth-order methods add perturbation during forward propagation, and afterward, the loss signal is passed to the middle layers. (e) The proposed OPZO method leverages momentum feedback connections based on perturbation vectors to directly propagate top-down error signals to neurons.

## 4    ONLINE PSEUDO-ZEROTH-ORDER TRAINING

In this section, we introduce the proposed online pseudo-zeroth-order method. We first introduce the pseudo-zeroth-order formulation for neural network training in Section 4.1. Then in Section 4.2, we introduce momentum feedback connections for error propagation with zeroth-order estimation of the model. In Section 4.3, we demonstrate the combination with online training and a similar form as the three-factor Hebbian learning. Finally, we introduce additional details in Section 4.4.

### 4.1    PSEUDO-ZEROTH-ORDER FORMULATION

Since zeroth-order methods suffer from large variances, a natural thought is to reduce the variance. However, ZO methods only rely on a scalar feedback signal to act on the random direction $\mathbf{z}$, making it hard to improve gradient estimation. To this end, we introduce a pseudo-zeroth-order formulation. As we build our work on online training, we first focus on the condition of a single SNN time step.

Specifically, we decouple the model function $f(\cdot; \boldsymbol{\theta})$ and the loss function $\mathcal{L}(\cdot)$. For each input $\mathbf{x}$, the model outputs $\mathbf{o} = f(\mathbf{x}; \boldsymbol{\theta})$, and then the loss is calculated as $\mathcal{L}(\mathbf{o}, \mathbf{y_x})$, where $\mathbf{y_x}$ is the label for the input. Different from ZO methods that only leverage the function value of $\mathcal{L} \circ f$, we assume that the gradient of $\mathcal{L}(\cdot)$ can be easily calculated, while keeping the zeroth-order formulation for $f(\cdot; \boldsymbol{\theta})$. This is consistent with real settings where gradients of the loss function have easy closed-form formulation, e.g., for mean-square-error (MSE) loss, $\nabla_{\mathbf{o}}\mathcal{L}(\mathbf{o}, \mathbf{y_x}) = \mathbf{o} - \mathbf{y_x}$, and for cross-entropy (CE) loss with the softmax function $\sigma$, $\nabla_{\mathbf{o}}\mathcal{L}(\mathbf{o}, \mathbf{y_x}) = \sigma(\mathbf{o}) - \mathbf{y_x}$, while gradients of $f(\cdot; \boldsymbol{\theta})$ are hard to compute due to biological plausibility issues or non-differentiability of spikes.

With this formulation, we can consider feedback (error) signals $\mathbf{e} = \nabla_{\mathbf{o}}\mathcal{L}(\mathbf{o}, \mathbf{y_x})$ that carries more information than a single value of $\mathcal{L} \circ f(\mathbf{x})$, potentially encouraging techniques for variance reduction. In the following, we introduce momentum feedback connections to directly propagate feedback signals to hidden layers for gradient estimation.

### 4.2    MOMENTUM FEEDBACK CONNECTIONS

We motivate our method by first considering the directional gradient by the two-point estimation in Section 3.2. With decoupled $f(\cdot; \boldsymbol{\theta})$ and $\mathcal{L}(\cdot)$ as in the pseudo-zeroth-order formulation and Taylor expansion of $\mathcal{L}(\cdot)$, the one-sided formulation turns into:

$$\nabla_{\boldsymbol{\theta}}^{ZO}\mathcal{L} \approx \frac{\langle \nabla_{\mathbf{o}}\mathcal{L}(\mathbf{o}, \mathbf{y_x}), \tilde{\mathbf{o}} - \mathbf{o}\rangle}{\alpha}\mathbf{z} = \mathbf{z}\frac{\Delta\mathbf{o}^\top}{\alpha}\nabla_{\mathbf{o}}\mathcal{L}(\mathbf{o}, \mathbf{y_x}), \tag{4}$$

where $\mathbf{o} = f(\mathbf{x}; \boldsymbol{\theta})$, $\tilde{\mathbf{o}} = f(\mathbf{x}; \boldsymbol{\theta} + \alpha\mathbf{z})$, and $\Delta\mathbf{o} = \tilde{\mathbf{o}} - \mathbf{o}$. This can be viewed as propagating the error signal with a connection weight $\mathbf{z}\frac{\Delta\mathbf{o}^\top}{\alpha}$. To reduce the variance introduced by the random direction $\mathbf{z}$,

we introduce momentum feedback connections across different iterations and propagate errors as:

$$\mathbf{M}^k := \lambda \mathbf{M}^{k-1} + (1-\lambda)\mathbf{z}\frac{\Delta\mathbf{o}^\top}{\alpha},$$
$$\nabla_{\boldsymbol{\theta}}^{PZO}\mathcal{L} = \mathbf{M}^k \nabla_{\mathbf{o}}\mathcal{L}(\mathbf{o}, \mathbf{y_x}), \tag{5}$$

where $\mathbf{M}$ is initialized as zero and $k$ denotes the iteration number. The momentum feedback connections can take advantage of different sampled directions $\mathbf{z}$, largely alleviating the variance caused by random directions.

The above formulation only considers the directional gradient with two-point estimation, while we are more interested in methods with a single forward pass. Actually, $\mathbf{z}\frac{\Delta\mathbf{o}^\top}{\alpha}$ can be viewed as a random estimator of $\mathbb{E}_{\mathbf{x}}\left[\mathbf{J}_f^\top(\mathbf{x})\right]$, where $\mathbf{J}_f(\mathbf{x})$ is the Jacobian of $f$ evaluated at $\mathbf{x}$, and $\mathbf{M}$ can be viewed as approximating it with moving average. Therefore, we can similarly use a one-point method:

$$\mathbf{M}^k := \lambda \mathbf{M}^{k-1} + (1-\lambda)\mathbf{z}\frac{\tilde{\mathbf{o}}^\top}{\alpha}, \tag{6}$$

where $\mathbf{z}\frac{\tilde{\mathbf{o}}^\top}{\alpha}$ is also an estimator of $\mathbb{E}_{\mathbf{x}}\left[\mathbf{J}_f^\top(\mathbf{x})\right]$, with the same expectation as $\mathbf{z}\frac{\Delta\mathbf{o}^\top}{\alpha}$ when $\alpha$ is given in practice. It is also an unbiased estimator of Jacobian of the smoothed version of $f$, not requiring $f$ to be differentiable (see Appendix A.1 for details).

This leads to our method as shown in Fig. 1(e). During forward propagation, a random noise $\alpha\mathbf{z}$ is injected for each layer, and momentum feedback connections are updated based on $\mathbf{z}$ and the model output $\tilde{\mathbf{o}}$ (information from pre- and post-synaptic neurons). Then errors are propagated through the connections to each layer. We consider node perturbation which is superior to weight perturbation[1] (Lillicrap et al., 2020), then it has a similar form as the popular DFA (Nøkland, 2016), while our feedback weight is not a random matrix but the estimated Jacobian (Fig. 1(c,e)).

Then we analyze some properties of momentum feedback connections. We assume that $\mathbf{M}$ can converge to the estimated $\mathbb{E}_{\mathbf{x}}\left[\mathbf{J}_f^\top(\mathbf{x})\right]$ up to small errors $\epsilon^2$, and we focus on gradient estimation with $\mathbf{M} = \mathbb{E}_{\mathbf{x}}\left[\mathbf{J}_f^\top(\mathbf{x})\right] + \epsilon$. We show that it largely reduces the variance of the zeroth-order method (the proof and discussions are in Appendix A).

**Proposition 4.1.** *Let $d$ denote the dimension of $\boldsymbol{\theta}$, $m$ denote the dimension of $\mathbf{o}$ ($m \ll d$), $B$ denote the mini-batch size, $\beta = \mathrm{Var}\left[z_i^2\right]$, $V_{\boldsymbol{\theta}} = \frac{1}{d}\sum_i \mathrm{Var}\left[(\nabla_{\boldsymbol{\theta}}\mathcal{L_x})_i\right]$, $S_{\boldsymbol{\theta}} = \frac{1}{d}\sum_i \mathbb{E}\left[(\nabla_{\boldsymbol{\theta}}\mathcal{L_x})_i\right]^2$, $V_L = \mathrm{Var}\left[\mathcal{L_x}\right]$, $S_L = \mathbb{E}\left[\mathcal{L_x}\right]^2$, $V_{\mathbf{o}} = \frac{1}{m}\sum_i \mathrm{Var}\left[(\nabla_{\mathbf{o}}\mathcal{L_x})_i\right]$, $S_{\mathbf{o}} = \frac{1}{m}\sum_i \mathbb{E}\left[(\nabla_{\mathbf{o}}\mathcal{L_x})_i\right]^2$, where $\mathcal{L_x}$ is the sample loss for input $\mathbf{x}$, and $\nabla_{\boldsymbol{\theta}}\mathcal{L_x}$ and $\nabla_{\mathbf{o}}\mathcal{L_x}$ are the sample gradient for $\boldsymbol{\theta}$ and $\mathbf{o}$, respectively. We further assume that the small error $\epsilon$ has i.i.d. components with zero mean and variance $V_{\epsilon}$, and let $V_{\mathbf{o},\mathbf{M}} = \frac{1}{d}\sum_{i,j} \mathrm{Var}\left[(\nabla_{\mathbf{o}}\mathcal{L_x})_j\right]\left(\mathbb{E}_{\mathbf{x}}\left[\mathbf{J}_f^\top(\mathbf{x})\right]\right)_{i,j}$. Then the average variance of the single-point zeroth-order method is: $\frac{1}{B}\left((d+\beta)V_{\boldsymbol{\theta}} + (d+\beta-1)S_{\boldsymbol{\theta}} + \frac{1}{\alpha^2}V_L + \frac{1}{\alpha^2}S_L\right) + O(\alpha^2)$, while that of the pseudo-zeroth-order method is: $\frac{1}{B}\left(mV_{\epsilon}V_{\mathbf{o}} + mV_{\epsilon}S_{\mathbf{o}} + V_{\mathbf{o},\mathbf{M}}\right)$.*

*Remark* 4.2. $V_{\boldsymbol{\theta}}$ corresponds to the sample variance of spatial BP, and $V_{\mathbf{o},\mathbf{M}}$ would be at a similar scale as $V_{\boldsymbol{\theta}}$ (see discussions in Appendix A). Since $V_{\epsilon}$ is expected to be very small, the results show that the zeroth-order estimation has at least $d$ times larger variance than BP, while the pseudo-zeroth-order method can significantly reduce the variance, which is also verified in experiments.

Besides the variance, another question is that momentum connections would take the expectation of the Jacobian over data $\mathbf{x}$, which can introduce bias into the gradient estimation. This is due to the data-dependent non-linearity that leads to a data-dependent Jacobian, which can be a shared problem for direct error feedback methods without layer-by-layer spatial BP. Despite the bias, we show that

---

[1]For $\mathbf{x}^{l+1} = \phi\left(\mathbf{W}^l\mathbf{x}^l\right)$, node perturbation estimates gradients for $\mathbf{x}^{l+1}$ and calculates gradients as $\nabla_{\mathbf{W}^l}\mathcal{L} = \left(\nabla_{\mathbf{x}^{l+1}}\mathcal{L} \odot \phi'\left(\mathbf{W}^l\mathbf{x}^l\right)\right)\mathbf{x}^{l\top}$, which has a smaller variance than directly estimating gradients for weights.

[2]If the parameters of the model are fixed, $\mathbf{M}$ is approximating a static matrix with projection to different directions, which can converge quickly. For the gradually evolving parameters, the expectation of the Jacobian over all samples may change slowly, and we can also expect $\mathbf{M}$ to track this expectation at a slow time scale.

under certain conditions, the estimated gradient can still provide a descent direction (the proof and discussions are in Appendix A).

**Proposition 4.3.** *Suppose that $\mathbf{J}_f^\top(\mathbf{x})$ is $L_J$-Liptschitz continuous and $\mathbf{e}(\mathbf{x})$ is $L_e$-Liptschitz continuous, $\mathbf{x}_i$ is uniformly distributed, when $\left\| \mathbb{E}_{\mathbf{x}_i} \left[ \mathbf{J}_f^\top(\mathbf{x}_i)\mathbf{e}(\mathbf{x}_i) \right] \right\| > \frac{1}{2} L_J L_e \Delta_{\mathbf{x}} + e_{\boldsymbol{\epsilon}}$, where $\Delta_{\mathbf{x}} = \mathbb{E}_{\mathbf{x}_i, \mathbf{x}_j} \left[ \|\mathbf{x}_i - \mathbf{x}_j\|^2 \right]$ and $e_{\boldsymbol{\epsilon}} = \|\boldsymbol{\epsilon} \mathbb{E}_{\mathbf{x}_i} [\mathbf{e}(\mathbf{x}_i)]\|$, we have $\left\langle \mathbb{E}_{\mathbf{x}_i} \left[ \mathbf{J}_f^\top(\mathbf{x}_i)\mathbf{e}(\mathbf{x}_i) \right], \mathbb{E}_{\mathbf{x}_i} [\mathbf{Me}(\mathbf{x}_i)] \right\rangle > 0$.*

Note that our analysis also holds for non-differentiable spiking neural networks. The single-point estimation is actually an unbiased estimator for the smoothed $f_\alpha$ with expectation over noise injection (Appendix A.1), where $f$ can be non-differentiable. This is similar to the stochastic setting where spiking neurons can be differentiable and gradients can be defined (see Appendix B for details). By treating $f$ in the analysis as $f_\alpha$, the analysis is effective.

## 4.3 ONLINE PSEUDO-ZEROTH-ORDER TRAINING

We build the above pseudo-zeroth-order approach on online training methods to deal with spatial and temporal credit assignments. As introduced in Section 3.1, we consider OTTT (Xiao et al., 2022) and replace its backpropagated instantaneous gradient with our estimated gradient based on direct top-down feedback. Then the update for synaptic weights has a similar form as the three-factor Hebbian learning (Frémaux & Gerstner, 2016), and the global modulator is a direct top-down signal without layer-by-layer BP:

$$\Delta W_{i,j} \propto \hat{a}_i[t]\psi(u_j[t]) \left( -g_j^t \right), \tag{7}$$

where $W_{i,j}$ is the weight from neuron $i$ to $j$, $\hat{a}_i[t]$ is the presynaptic activity trace, $\psi(u_j[t])$ is a local surrogate derivative for the change rate of the postsynaptic activity (Xiao et al., 2022), and $g_j^t$ is the global top-down error (gradient) modulator. Here we leverage the local surrogate derivative because it can be well-defined under the stochastic setting (see Appendix B) and better fits the biological rule.

For potentially asynchronous neuromorphic computing, there may be a delay in the propagation of error signals. Xiao et al. (2022) show that with convergent inputs and certain surrogate derivatives, the gradient is still theoretically effective under the delay $\Delta t$, i.e., the update is based on $\hat{a}_i[t + \Delta t]\psi(u_j[t + \Delta t])g_j^t$. Alternatively, more eligibility traces can be used to store the local information, e.g., $\hat{a}_i[t]\psi(u_j[t])$, and induce weight updates when the top-down signal arrives (Bellec et al., 2020). Our method shares these properties and we do not model delays in experiments for efficiency.

Moreover, the direct error propagations to different layers as well as the update of feedback connections in our method can be parallel, which can better take advantage of parallel neuromorphic computing than layer-by-layer spatial BP.

## 4.4 ADDITIONAL DETAILS

**Combination with Local Learning**  There can be both global and local signals for learning in biological systems, and local learning (LL) can improve global learning approximation methods (Ren et al., 2023). Our proposed method can be combined with LL as well. We consider introducing local readout layers, where a fully connected readout is added for each layer with supervised loss. Additionally, we can also introduce intermediate global learning (IGL) that propagates global signals from a middle layer to previous ones with OPZO. More details can be found in Appendix C.

**About Noise Injection**  By default, we sample $\mathbf{z}$ from the Gaussian distribution. As sampling from the Gaussian distribution may pose computational requirements for hardware, we can also consider easier distributions such as the Rademacher distribution, which takes $1$ and $-1$ both with the probability $0.5$. Sampling from unit spheres is also feasible. Additionally, $\mathbf{z}$ is by default added to the neural activities for gradient estimation based on node perturbation. To further prevent perturbation from interfering with sparse spike-driven forward propagation, we may empirically change the noise injection as perturbation before neurons (i.e., perturb on membrane potentials), while maintaining local surrogate derivatives for the spiking function. We will show in experiments that OPZO is robust to these noise injection settings. Additionally, we can leverage antithetic $\mathbf{z}$, i.e., $\mathbf{z}$ and $-\mathbf{z}$, for every two time steps of SNNs to further reduce the variance. More details can be found in Appendix C.

Table 1: Accuracy (%) of different spatial credit assignment methods with online training on various datasets.

| Method | N-MNIST | DVS-Gesture | DVS-CIFAR10 | MNIST | CIFAR-10 | CIFAR-100 |
|---|---|---|---|---|---|---|
| BP (Xiao et al., 2022) | 98.15±0.05 | 95.72±0.33 | 75.43±0.39 | **98.38±0.02** | **90.00±0.06** | 64.82±0.09 |
| BP (w/ LL) | / | 95.72±0.71 | 76.07±0.41 | / | 89.82±0.16 | **64.88±0.08** |
| DFA (Nøkland, 2016) | 97.98±0.03 | 91.67±0.75 | 60.60±0.67 | 98.05±0.04 | 79.90±0.15 | 49.50±0.13 |
| DFA (w/ LL) | / | 91.43±0.59 | 61.77±0.62 | / | 82.38±0.22 | 54.76±0.21 |
| DKP (Webster et al., 2020) | 97.87±0.05 | 60.53±7.82 | 37.70±1.21 | 98.15±0.03 | 81.84±0.96 | 53.27±0.34 |
| ZO$_{sp}$ | 72.90±1.14 | 23.73±2.38 | 31.67±0.24 | 86.53±0.11 | 49.04±0.63 | 22.26±0.51 |
| OPZO | **98.27±0.04** | 94.33±0.16 | 72.77±0.82 | 98.34±0.10 | 85.74±0.15 | 60.93±0.16 |
| OPZO (w/ LL) | / | **96.06±0.33** | **77.47±0.12** | / | 89.42±0.16 | 64.77±0.16 |

Figure 2: Results of gradient variances of OPZO, spatial BP, and ZO$_{sp}$ on different datasets. "L$i$" denotes the $i$-th layer.

## 5 EXPERIMENTS

In this section, we conduct experiments on both neuromorphic and static datasets with fully connected (FC) and convolutional (Conv) neural networks to demonstrate the effectiveness of the proposed OPZO method. For N-MNIST and MNIST, we leverage FC networks with two hidden layers composed of 800 neurons, and for DVS-CIFAR10, DVS-Gesture, CIFAR-10, and CIFAR-100, we leverage 5-layer convolutional networks. We will also consider a deeper 9-layer convolutional network, as well as fine-tuning ResNet-34 on ImageNet under noise. We take $T = 30$ time steps for N-MNIST, $T = 20$ for DVS-Gesture, $T = 10$ for DVS-CIFAR10, and $T = 6$ time steps for static datasets, following previous works (Xiao et al., 2022; Zhang & Li, 2020). More training details can be found in Appendix C.

### 5.1 COMPARISON ON VARIOUS DATASETS

We first compare the proposed OPZO with other spatial credit assignment methods on various datasets in Table 1, and all methods are based on the online training method OTTT (Xiao et al., 2022) under the same settings. The compared methods include spatial BP, DFA (Nøkland, 2016), DKP (Webster et al., 2020) that learns feedback connections in DFA, single-point zeroth-order method, and the combination with local learning. We do not consider local learning settings for FC networks since there are only two hidden layers. As shown in the results, the ZO$_{sp}$ method fails to effectively optimize neural networks, while OPZO significantly improves the results, achieving performance at a similar level as spatial BP. DFA with random feedback has a large gap with spatial BP, especially on convolutional networks, while OPZO can achieve much better results. DKP improves DFA on static datasets, but it performs poorly on neuromorphic datasets and has significant gaps with OPZO on all datasets. When combined with local learning, OPZO (w/ LL) has about the same performance as BP (w/ LL) and even outperforms BP (w/ LL) on neuromorphic datasets. These results demonstrate the effectiveness of OPZO for promising performance in a more biologically plausible and neuromorphic-friendly approach, paving paths for direct on-chip training of SNNs.

Note that our method is a different line from most recent works with state-of-the-art performance (Li et al., 2023; Zhou et al., 2023; Guo et al., 2024; Yao et al., 2024), which are based on spatio-temporal BP and focus on architecture or training objective improvement. We aim to develop alternatives to BP, focusing on more biologically plausible and hardware-friendly training algorithms. So we mainly compare different spatial credit assignment methods under the same settings.

Table 2: Accuracy (%) of OPZO on CIFAR-10 with different kinds of noise injection.

| Distribution | Pert. after neuron | Pert. before neuron |
|---|---|---|
| Gaussian | 85.73±0.15 | 84.37±0.13 |
| Unit Sphere | 86.01±0.28 | 84.50±0.13 |
| Rademacher | 85.69±0.17 | 84.03±0.23 |

Table 3: Accuracy (%) of different methods with a deeper network.

| Method | DVS-Gesture | CIFAR-100 |
|---|---|---|
| Spatial BP | 94.10±1.02 | 65.96±0.52 |
| DFA (w/ LL) | 93.40±0.49 | 52.94±0.20 |
| DFA (w/ LL&IGL) | 93.29±0.33 | 54.17±0.54 |
| OPZO (w/ LL) | 95.83±0.85 | 65.87±0.13 |
| OPZO (w/ LL&IGL) | **96.88±0.28** | **66.13±0.15** |

## 5.2 Gradient Variance

We analyze the gradient variance of different methods to verify that our method can effectively reduce variance for effective training. As shown in Fig. 2, the variance of $ZO_{sp}$ is several orders larger than spatial BP, leading to the failure of effective training. OPZO can largely reduce the variance to have a similar scale as BP, which is consistent with our theoretical analysis.

## 5.3 Effectiveness for Different Noise Injection

Then we verify the effectiveness of OPZO for different noise injection settings as introduced in Section 4.4. As shown in Table 2, the results under different noise distributions and injection positions are similar, demonstrating the robustness of OPZO for different settings.

## 5.4 Deeper Networks

We further consider deeper and larger networks. We first perform experiments with a deeper 9-layer convolutional network. We leverage local learning and intermediate global learning (Section 4.4). As shown in Table 3, OPZO can also achieve similar performance as or outperform spatial BP and significantly outperform DFA combined with these techniques. We also analyze pure OPZO without auxiliary techniques and its scalability to deeper networks with residual connections in Section D.5, showing that pure OPZO has more reliance on the proper network structure (residual connections) than BP for depth scalability.

We also conduct experiments for fine-tuning ResNet-34 on ImageNet under noise. This task is on the ground that there can be hardware mismatch, e.g., hardware noise, for deploying SNNs to neuromorphic hardware (Yang et al., 2022; Cramer et al., 2022), and we may expect direct on-chip fine-tuning to better deal with the problem. Our method is more plausible and efficient for on-chip learning than spatial BP and may be combined with other works aiming at high-performance training on common devices in this scenario. We fine-tune a pre-trained NF-ResNet-34 model released by Xiao et al. (2022) (original test accuracy 65.15%) under the noise injection setting with different scales. As shown in Table 4, OPZO can successfully fine-tune the model, while DFA and $ZO_{sp}$ fail. Spatial BP is neuromorphic-unfriendly, so its results are only for reference. The results show that OPZO can scale to large-scale settings.

Table 4: Accuracy (%) of different methods for fine-tuning ResNet-34 on ImageNet under different noise scale (n.s.). "Test" refers to the direct test of the original model. "BP" refers to spatial BP.

| **ImageNet** | | | | | |
|---|---|---|---|---|---|
| n.s. | Test | BP | DFA | $ZO_{sp}$ | OPZO |
| 0.1 | 61.13 | 63.91 | 61.20 | 52.42 | 63.39 |
| 0.15 | 54.01 | 62.13 | 54.59 | 30.32 | 60.96 |

Table 5: Estimation of training costs on potential neuromorphic hardware for $N$-hidden-layer neural networks ($n$ neurons for hidden layers, $m$ neurons for the output, $m \ll n$). The costs focus on the error backward procedure. "*" denotes parallelizable for different layers.

| Method | Memory | Operations |
|---|---|---|
| BP (if possible) | $O\left((N-1)n^2 + mn\right)$ | $O\left((N-1)n^2 + mn\right)$ |
| DFA* | $O\left(Nmn\right)$ | $O\left(Nmn\right)$ |
| $ZO_{sp}$* | $O(Nn)$ | $O(Nn)$ |
| OPZO* | $O\left(Nmn\right)$ | $O\left(Nmn\right)$ |

## 5.5 Training Costs and Firing Sparsity

Finally, we analyze and compare the computational costs of different methods. We mainly consider the estimated costs on potential neuromorphic hardware, which is the target of SNNs. Since biological systems leverage unidirectional local synapses, spatial BP (if assuming possible for weight transport and separate forward-backward stage) should maintain additional backward layer-by-layer connections for error backpropagation, leading to high memory and operation costs, as shown in Table 5. Differently, DFA and OPZO maintain direct top-down feedback with much smaller costs, which are also parallelizable for different layers. $ZO_{sp}$ may have even lower costs by propagating only a scalar signal, but it is ineffective in practice. Also note that, different from previous zeroth-order methods that require multiple forward propagations, our method only needs one common forward propagation with noise injection and direct top-down feedback, keeping lower operation costs similar to DFA. We also provide training costs on GPU in Appendix D, and our method is comparable to spatial BP and DFA, since GPUs do not follow neuromorphic properties. It is interesting future work to consider applications to neuromorphic hardware that is still under development (Davies, 2021; Schuman et al., 2022).

We further study the firing rate and synaptic operations of the trained models in Appendix D.2, showing that models trained by OPZO combined with local learning achieve the lowest operations, i.e., the most energy efficient. Additionally, we perform more analysis experiments of hyperparameters in Section D.4. Please refer to Appendix D for more details and results.

## 6 Conclusion

In this work, we propose the online pseudo-zeroth-order method for training spiking neural networks in a more biologically plausible and neuromorphic-hardware-friendly way, with low estimated costs and competitive performance compared with spatial BP. OPZO performs spatial credit assignment by a single forward propagation with noise injection and direct top-down feedback with momentum feedback connections, avoiding drawbacks of spatial BP, solving the large variance problem of zeroth-order methods, and significantly outperforming random feedback methods. With online training, OPZO has a similar form as three-factor Hebbian learning with direct top-down modulations, taking a step forward towards on-chip SNN training. Extensive experiments demonstrate the effectiveness and robustness of OPZO for both fully connected and convolutional networks on neuromorphic and static datasets.

### Acknowledgments

Z. Lin was supported by the NSF China (No. 62276004).

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

## A    Detailed Explanations and Proofs

In this section, we provide more explanations and proofs for propositions in the main text.

### A.1    Explanation of the smoothed function

For the neural network function $f$ with node perturbation given scale $\alpha$, the smoothed version of $f$ is defined as $f_\alpha(\cdot; \theta) = \mathbb{E}_{\mathbf{z}}[\hat{f}(\cdot; \theta, \alpha \mathbf{z})]$, where $\hat{f}$ refers to injecting noise $\alpha \mathbf{z}$ for node perturbation. Similar to Flaxman et al. (2005), by extending the gradient to the Jacobian, we can show that the one-point formulation of $\mathbf{z}\frac{\tilde{\mathbf{o}}^\top}{\alpha}$ is an unbiased estimator of the Jacobian of $f_\alpha$.

**Lemma A.1.** *When $\mathbf{z}$ is uniformly sampled from the unit sphere, $\mathbf{z}\frac{\tilde{\mathbf{o}}^\top}{\alpha}$ is an unbiased estimator of $\mathbf{J}_{f_\alpha}^\top(\mathbf{x})$ given $\mathbf{x}$, and further, are unbiased estimators of $\mathbb{E}_{\mathbf{x}}\left[\mathbf{J}_{f_\alpha}^\top(\mathbf{x})\right]$.*

### A.2    Proof of Proposition 4.1

*Proof.* We first consider the average variance of the two-point ZO estimation $\nabla_{\boldsymbol{\theta}}^{ZO}\mathcal{L} = \mathbf{z}\mathbf{z}^\top \nabla_{\boldsymbol{\theta}}\mathcal{L} + O(\alpha)$. Since $\text{Var}(xy) = \text{Var}(x)\text{Var}(y) + \text{Var}(x)\mathbb{E}(y)^2 + \text{Var}(y)\mathbb{E}(x)^2$ for independent $x$ and $y$, and $\mathbb{E}[z_i^2] = \text{Var}[z_i] + \mathbb{E}[z_i]^2 = 1$, for each element of the gradient under sample $\mathbf{x}$, we have:

$$\text{Var}\left[(\nabla_{\boldsymbol{\theta}}^{ZO}\mathcal{L}_{\mathbf{x}})_i\right]$$

$$= \text{Var}\left[\sum_{j=1}^{d} z_i z_j (\nabla_{\boldsymbol{\theta}}\mathcal{L}_{\mathbf{x}})_j\right] + O(\alpha^2)$$

$$= \text{Var}\left[z_i^2 (\nabla_{\boldsymbol{\theta}}\mathcal{L}_{\mathbf{x}})_i\right] + \sum_{j\neq i}\text{Var}\left[z_i z_j (\nabla_{\boldsymbol{\theta}}\mathcal{L}_{\mathbf{x}})_j\right] + O(\alpha^2)$$

$$= \text{Var}\left[z_i^2\right]\text{Var}\left[(\nabla_{\boldsymbol{\theta}}\mathcal{L}_{\mathbf{x}})_i\right] + \text{Var}\left[z_i^2\right]\mathbb{E}\left[(\nabla_{\boldsymbol{\theta}}\mathcal{L}_{\mathbf{x}})_i\right]^2 + \text{Var}\left[(\nabla_{\boldsymbol{\theta}}\mathcal{L}_{\mathbf{x}})_i\right]\mathbb{E}\left[z_i^2\right]^2$$

$$+ \sum_{j\neq i}\left(\text{Var}\left[z_i z_j\right]\text{Var}\left[(\nabla_{\boldsymbol{\theta}}\mathcal{L}_{\mathbf{x}})_j\right] + \text{Var}\left[z_i z_j\right]\mathbb{E}\left[(\nabla_{\boldsymbol{\theta}}\mathcal{L}_{\mathbf{x}})_j\right]^2 + \text{Var}\left[(\nabla_{\boldsymbol{\theta}}\mathcal{L}_{\mathbf{x}})_j\right]\mathbb{E}\left[z_i z_j\right]^2\right) + O(\alpha^2)$$

$$= (\beta + 1)\text{Var}\left[(\nabla_{\boldsymbol{\theta}}\mathcal{L}_{\mathbf{x}})_i\right] + \beta\mathbb{E}\left[(\nabla_{\boldsymbol{\theta}}\mathcal{L}_{\mathbf{x}})_i\right]^2 + \sum_{j\neq i}\left(\text{Var}\left[(\nabla_{\boldsymbol{\theta}}\mathcal{L}_{\mathbf{x}})_j\right] + \mathbb{E}\left[(\nabla_{\boldsymbol{\theta}}\mathcal{L}_{\mathbf{x}})_j\right]^2\right) + O(\alpha^2)$$

$$= \beta\text{Var}\left[(\nabla_{\boldsymbol{\theta}}\mathcal{L}_{\mathbf{x}})_i\right] + (\beta - 1)\mathbb{E}\left[(\nabla_{\boldsymbol{\theta}}\mathcal{L}_{\mathbf{x}})_i\right]^2 + \sum_{j=1}^{d}\left(\text{Var}\left[(\nabla_{\boldsymbol{\theta}}\mathcal{L}_{\mathbf{x}})_j\right] + \mathbb{E}\left[(\nabla_{\boldsymbol{\theta}}\mathcal{L}_{\mathbf{x}})_j\right]^2\right) + O(\alpha^2).$$

$$(8)$$

Taking the average of all elements, we obtain the average variance for each sample (denoted as mVar):

$$\text{mVar}\left[\nabla_{\boldsymbol{\theta}}^{ZO}\mathcal{L}_{\mathbf{x}}\right]$$

$$= \frac{1}{d}\sum_{i=1}^{d}\text{Var}\left[(\nabla_{\boldsymbol{\theta}}^{ZO}\mathcal{L}_{\mathbf{x}})_i\right]$$

$$= \frac{\beta}{d}\sum_{i=1}^{d}\text{Var}\left[(\nabla_{\boldsymbol{\theta}}\mathcal{L}_{\mathbf{x}})_i\right] + \frac{\beta - 1}{d}\sum_{i=1}^{d}\mathbb{E}\left[(\nabla_{\boldsymbol{\theta}}\mathcal{L}_{\mathbf{x}})_i\right]^2 + \sum_{j=1}^{d}\left(\text{Var}\left[(\nabla_{\boldsymbol{\theta}}\mathcal{L}_{\mathbf{x}})_j\right] + \mathbb{E}\left[(\nabla_{\boldsymbol{\theta}}\mathcal{L}_{\mathbf{x}})_j\right]^2\right) + O(\alpha^2)$$

$$= (d + \beta)V_{\boldsymbol{\theta}} + (d + \beta - 1)S_{\boldsymbol{\theta}} + O(\alpha^2).$$

$$(9)$$

For gradient calculation with batch size $B$, the sample variance can be reduced by $B$ times, resulting in the average variance $\frac{1}{B}\left((d + \beta)V_{\boldsymbol{\theta}} + (d + \beta - 1)S_{\boldsymbol{\theta}}\right) + O(\alpha^2)$.

Then we can derive the average variance of the single-point ZO estimation $\nabla_{\boldsymbol{\theta}}^{ZO_{sp}} \mathcal{L} = \nabla_{\boldsymbol{\theta}}^{ZO} \mathcal{L} + \frac{\mathcal{L}_{\mathbf{x}}}{\alpha} \mathbf{z}$ for each sample:

$$
\begin{aligned}
& \mathrm{mVar}\left[\nabla_{\boldsymbol{\theta}}^{ZO_{sp}} \mathcal{L}_{\mathbf{x}}\right] \\
&= \mathrm{mVar}\left[\nabla_{\boldsymbol{\theta}}^{ZO} \mathcal{L}_{\mathbf{x}}\right] + \mathrm{mVar}\left[\frac{\mathcal{L}_{\mathbf{x}}}{\alpha} \mathbf{z}\right] \\
&= (d+\beta)V_{\boldsymbol{\theta}} + (d+\beta-1)S_{\boldsymbol{\theta}} + O(\alpha^2) + \frac{1}{\alpha^2}\left(\mathrm{Var}\left[\mathcal{L}_{\mathbf{x}}\right]\mathrm{Var}\left[z_i\right] + \mathrm{Var}\left[\mathcal{L}_{\mathbf{x}}\right]\mathbb{E}\left[z_i\right]^2 + \mathrm{Var}\left[z_i\right]\mathbb{E}\left[\mathcal{L}_{\mathbf{x}}\right]^2\right) \\
&= (d+\beta)V_{\boldsymbol{\theta}} + (d+\beta-1)S_{\boldsymbol{\theta}} + \frac{1}{\alpha^2}V_L + \frac{1}{\alpha^2}S_L + O(\alpha^2).
\end{aligned}
\tag{10}
$$

For batch size $B$, the average variance is $\frac{1}{B}\left((d+\beta)V_{\boldsymbol{\theta}} + (d+\beta-1)S_{\boldsymbol{\theta}} + \frac{1}{\alpha^2}V_L + \frac{1}{\alpha^2}S_L\right) + O(\alpha^2)$.

Next, we turn to the average variance of the pseudo-zeroth-order method $\nabla_{\boldsymbol{\theta}}^{PZO} \mathcal{L} = \mathbf{M}\nabla_{\mathbf{o}}\mathcal{L}_{\mathbf{x}} = \left(\mathbb{E}_{\mathbf{x}}\left[\mathbf{J}_f^\top(\mathbf{x})\right] + \boldsymbol{\epsilon}\right)\nabla_{\mathbf{o}}\mathcal{L}_{\mathbf{x}}$. For each element, we have:

$$
\begin{aligned}
& \mathrm{Var}\left[\left(\nabla_{\boldsymbol{\theta}}^{PZO}\mathcal{L}_{\mathbf{x}}\right)_i\right] \\
&= \mathrm{Var}\left[\sum_{j=1}^{m}\left(\mathbb{E}_{\mathbf{x}}\left[\mathbf{J}_f^\top(\mathbf{x})\right]\right)_{i,j}(\nabla_{\mathbf{o}}\mathcal{L}_{\mathbf{x}})_j\right] + \mathrm{Var}\left[\sum_{j=1}^{m}\epsilon_{i,j}(\nabla_{\mathbf{o}}\mathcal{L}_{\mathbf{x}})_j\right] \\
&= \sum_{j=1}^{m}\left(\mathbb{E}_{\mathbf{x}}\left[\mathbf{J}_f^\top(\mathbf{x})\right]\right)_{i,j}\mathrm{Var}\left[(\nabla_{\mathbf{o}}\mathcal{L}_{\mathbf{x}})_j\right] + \sum_{j=1}^{m}\left(V_{\epsilon}\mathrm{Var}\left[(\nabla_{\mathbf{o}}\mathcal{L}_{\mathbf{x}})_j\right] + V_{\epsilon}\mathbb{E}\left[(\nabla_{\mathbf{o}}\mathcal{L}_{\mathbf{x}})_j\right]^2\right).
\end{aligned}
\tag{11}
$$

Taking the average of all elements, we have the average variance for each sample:

$$
\begin{aligned}
& \mathrm{mVar}\left[\nabla_{\boldsymbol{\theta}}^{PZO}\mathcal{L}_{\mathbf{x}}\right] \\
&= \frac{1}{d}\sum_{i=1}^{d}\sum_{j=1}^{m}\left(\mathbb{E}_{\mathbf{x}}\left[\mathbf{J}_f^\top(\mathbf{x})\right]\right)_{i,j}\mathrm{Var}\left[(\nabla_{\mathbf{o}}\mathcal{L}_{\mathbf{x}})_j\right] + \sum_{j=1}^{m}\left(V_{\epsilon}\mathrm{Var}\left[(\nabla_{\mathbf{o}}\mathcal{L}_{\mathbf{x}})_j\right] + V_{\epsilon}\mathbb{E}\left[(\nabla_{\mathbf{o}}\mathcal{L}_{\mathbf{x}})_j\right]^2\right) \\
&= mV_{\epsilon}V_{\mathbf{o}} + mV_{\epsilon}S_{\mathbf{o}} + V_{\mathbf{o},\mathbf{M}}.
\end{aligned}
\tag{12}
$$

Then for batch size $B$, the average variance is $\frac{1}{B}\left(mV_{\epsilon}V_{\mathbf{o}} + mV_{\epsilon}S_{\mathbf{o}} + V_{\mathbf{o},\mathbf{M}}\right)$. $\qquad\square$

*Remark* A.2. $\beta = \mathrm{Var}\left[z_i^2\right] = \mathbb{E}(z_i^4) - \mathbb{E}(z_i^2)^2 = \mathbb{E}(z_i^4) - 1$ depends on the distribution of $z_i$. For the Gaussian distribution, $\mathbb{E}(z_i^4) = 3$ and therefore $\beta = 2$. For the Rademacher distribution, $\mathbb{E}(z_i^4) = 1$ and therefore $\beta = 0$.

*Remark* A.3. The zero mean assumption on the small error $\boldsymbol{\epsilon}$ is reasonable, when we actually consider $f_\alpha$ and $\mathbf{z}\frac{\tilde{\mathbf{o}}^\top}{\alpha}$ is an unbiased estimator for $\mathbb{E}_{\mathbf{x}}\left[\mathbf{J}_{f_\alpha}^\top(\mathbf{x})\right]$ (Lemma A.1), so the expectation of the error can be expected to be zero.

*Remark* A.4. $V_{\boldsymbol{\theta}}$ and $V_{\mathbf{o},\mathbf{M}}$ may not be directly compared considering the complex network function, but we may make a brief analysis under some simplifications. For $(\nabla_{\boldsymbol{\theta}}\mathcal{L}_{\mathbf{x}})_i = \left(\mathbf{J}_f^\top(\mathbf{x})\nabla_{\mathbf{o}}\mathcal{L}_{\mathbf{x}}\right)_i$, let

$\mathbf{J}_{i,j}$ and $\nabla_j$ denote $\left(\mathbf{J}_f^\top(\mathbf{x})\right)_{i,j}$ and $(\nabla_\mathbf{o}\mathcal{L}_\mathbf{x})_j$ for short, we have

$$\mathrm{Var}\left[(\nabla_{\boldsymbol{\theta}}\mathcal{L}_\mathbf{x})_i\right] = \mathrm{Var}\left[\sum_{j=1}^m \mathbf{J}_{i,j}\nabla_j\right] = \sum_j \mathrm{Var}\left[\mathbf{J}_{i,j}\nabla_j\right] + \sum_{j_1,j_2}\mathrm{Cov}\left[\mathbf{J}_{i,j_1}\nabla_{j_1}, \mathbf{J}_{i,j_2}\nabla_{j_2}\right]$$

$$= \sum_j\left[\mathrm{Var}\left[\nabla_j\right]\mathbb{E}\left[\mathbf{J}_{i,j}^2\right] + \mathrm{Var}\left[\mathbf{J}_{i,j}\right]\mathbb{E}\left[\nabla_j\right]^2 + 2\mathrm{Cov}\left[\mathbf{J}_{i,j}, \nabla_j\right]\mathbb{E}\left[\mathbf{J}_{i,j}\right]\mathbb{E}\left[\nabla_j\right]\right] + \sum_{j_1,j_2}\mathrm{Cov}\left[\mathbf{J}_{i,j_1}\nabla_{j_1}, \mathbf{J}_{i,j_2}\nabla_{j_2}\right].$$

(13)

If we ignore covariance terms and assume $\mathbb{E}\left[\nabla_j\right] = 0$, this is simplified to $\sum_j \mathrm{Var}\left[\nabla_j\right]\mathbb{E}\left[\mathbf{J}_{i,j}^2\right]$, and then $V_{\boldsymbol{\theta}}$ is approximated as $\frac{1}{d}\sum_{i,j}\mathrm{Var}\left[\nabla_j\right]\mathbb{E}\left[\mathbf{J}_{i,j}^2\right]$, which has a similar form as $V_{\mathbf{o},\mathbf{M}} = \frac{1}{d}\sum_{i,j}\mathrm{Var}\left[(\nabla_\mathbf{o}\mathcal{L}_\mathbf{x})_j\right]\left(\mathbb{E}_\mathbf{x}\left[\mathbf{J}_f^\top(\mathbf{x})\right]\right)_{i,j}$ except that the second moment is considered. Under this condition, the scales of $V_{\mathbf{o},\mathbf{M}}$ and $V_{\boldsymbol{\theta}}$ may slightly differ considering the scale of elements of $\mathbf{J}_f^\top(\mathbf{x})$, but overall, $V_{\mathbf{o},\mathbf{M}}$ would be at a similar scale as $V_{\boldsymbol{\theta}}$ compared with the variances of the zeroth-order methods that are at least $d$ times larger which is proportional to the number of intermediate neurons.

### A.3 PROOF OF PROPOSITION 4.3

*Proof.* Since $\mathbf{J}_f^\top(\mathbf{x})$ is $L_J$-Liptschitz continuous and $\mathbf{e}(\mathbf{x})$ is $L_e$-Liptschitz continuous, we have $\left\|\mathbf{J}_f^\top(\mathbf{x}_i) - \mathbf{J}_f^\top(\mathbf{x}_j)\right\| \le L_J\|\mathbf{x}_i - \mathbf{x}_j\|$, $\|\mathbf{e}(\mathbf{x}_i) - \mathbf{e}(\mathbf{x}_j)\| \le L_e\|\mathbf{x}_i - \mathbf{x}_j\|$. Then with the equation that $\frac{1}{2n^2}\sum_{i,j}(a_i - a_j)(b_i - b_j) = \frac{1}{n}\sum_i a_i b_i - \frac{1}{n^2}\sum_{i,j}a_i b_j$, we have

$$\left\|\mathbb{E}_{\mathbf{x}_i}\left[\mathbf{J}_f^\top(\mathbf{x}_i)\mathbf{e}(\mathbf{x}_i)\right] - \mathbb{E}_{\mathbf{x}_i}\left[\left(\mathbb{E}_{\mathbf{x}_j}\left[\mathbf{J}_f^\top(\mathbf{x}_j)\right] + \boldsymbol{\epsilon}\right)\mathbf{e}(\mathbf{x}_i)\right]\right\|$$

$$= \left\|\frac{1}{n}\sum_{\mathbf{x}_i}\widetilde{\mathbf{J}_\mathbf{f}}(\mathbf{x}_i)\mathbf{e}(\mathbf{x}_i) - \left(\frac{1}{n}\sum_{\mathbf{x}_i}\widetilde{\mathbf{J}_\mathbf{f}}(\mathbf{x}_i)\right)\left(\frac{1}{n}\sum_{\mathbf{x}_i}\mathbf{e}(\mathbf{x}_i)\right) - \boldsymbol{\epsilon}\mathbb{E}_{\mathbf{x}_i}\left[\mathbf{e}(\mathbf{x}_i)\right]\right\|$$

$$= \left\|\frac{1}{2n^2}\sum_{\mathbf{x}_i,\mathbf{x}_j}\left(\widetilde{\mathbf{J}_\mathbf{f}}(\mathbf{x}_i) - \widetilde{\mathbf{J}_\mathbf{f}}(\mathbf{x}_j)\right)(\mathbf{e}(\mathbf{x}_i) - \mathbf{e}(\mathbf{x}_j)) - \boldsymbol{\epsilon}\mathbb{E}_{\mathbf{x}_i}\left[\mathbf{e}(\mathbf{x}_i)\right]\right\|$$

$$\le \frac{1}{2n^2}\sum_{\mathbf{x}_i,\mathbf{x}_j}\left\|\left(\widetilde{\mathbf{J}_\mathbf{f}}(\mathbf{x}_i) - \widetilde{\mathbf{J}_\mathbf{f}}(\mathbf{x}_j)\right)\right\|\|(\mathbf{e}(\mathbf{x}_i) - \mathbf{e}(\mathbf{x}_j))\| + \|\boldsymbol{\epsilon}\mathbb{E}_{\mathbf{x}_i}\left[\mathbf{e}(\mathbf{x}_i)\right]\|$$ (14)

$$\le \frac{1}{2n^2}\sum_{\mathbf{x}_i,\mathbf{x}_j}L_J L_e\|\mathbf{x}_i - \mathbf{x}_j\|^2 + \|\boldsymbol{\epsilon}\mathbb{E}_{\mathbf{x}_i}\left[\mathbf{e}(\mathbf{x}_i)\right]\|$$

$$= \frac{1}{2}L_J L_e\Delta_\mathbf{x} + e_{\boldsymbol{\epsilon}}$$

$$< \left\|\mathbb{E}_{\mathbf{x}_i}\left[\mathbf{J}_f^\top(\mathbf{x}_i)\mathbf{e}(\mathbf{x}_i)\right]\right\|.$$

Therefore,

$$\left\langle\mathbb{E}_{\mathbf{x}_i}\left[\mathbf{J}_f^\top(\mathbf{x}_i)\mathbf{e}(\mathbf{x}_i)\right], \mathbb{E}_{\mathbf{x}_i}\left[\mathbf{M}\mathbf{e}(\mathbf{x}_i)\right]\right\rangle$$

$$= \left\|\mathbb{E}_{\mathbf{x}_i}\left[\mathbf{J}_f^\top(\mathbf{x}_i)\mathbf{e}(\mathbf{x}_i)\right]\right\|^2 - \left\langle\mathbb{E}_{\mathbf{x}_i}\left[\mathbf{J}_f^\top(\mathbf{x}_i)\mathbf{e}(\mathbf{x}_i)\right], \mathbb{E}_{\mathbf{x}_i}\left[\mathbf{J}_f^\top(\mathbf{x}_i)\mathbf{e}(\mathbf{x}_i)\right] - \mathbb{E}_{\mathbf{x}_i}\left[\mathbf{M}\mathbf{e}(\mathbf{x}_i)\right]\right\rangle$$

$$\ge \left\|\mathbb{E}_{\mathbf{x}_i}\left[\mathbf{J}_f^\top(\mathbf{x}_i)\mathbf{e}(\mathbf{x}_i)\right]\right\|^2 - \left\|\mathbb{E}_{\mathbf{x}_i}\left[\mathbf{J}_f^\top(\mathbf{x}_i)\mathbf{e}(\mathbf{x}_i)\right]\right\|\left\|\mathbb{E}_{\mathbf{x}_i}\left[\mathbf{J}_f^\top(\mathbf{x}_i)\mathbf{e}(\mathbf{x}_i)\right] - \mathbb{E}_{\mathbf{x}_i}\left[\left(\mathbb{E}_{\mathbf{x}_j}\left[\mathbf{J}_f^\top(\mathbf{x}_j)\right] + \boldsymbol{\epsilon}\right)\mathbf{e}(\mathbf{x}_i)\right]\right\|$$

$$> 0.$$

(15)

$\square$

*Remark A.5.* $L_J$ will depend on the smoothness of the network, for example, $L_J = 0$ for linear networks. This will influence the condition of effective descent direction considering the gradient scale as in the proposition. Note that these assumptions are not necessary premises, and we have verified the effectiveness of the method in experiments.

# B   INTRODUCTION TO LOCAL SURROGATE DERIVATIVES UNDER THE STOCHASTIC SPIKING SETTING

In this section, we provide more introduction to the stochastic spiking setting, under which spiking neurons can be *locally* differentiable and there exist *local* surrogate derivatives.

Biological spiking neurons can be stochastic, where a neuron generates spikes following a Bernoulli distribution with the probability as the c.d.f. of a distribution w.r.t $u[t] - V_{th}$, indicating a higher probability for a spike with larger $u[t] - V_{th}$. That is, $s_i[t]$ is a random variable following a $\{0, 1\}$ valued Bernoulli distribution with the probability of 1 as $p(s_i[t] = 1) = F(u_i[t] - V_{th})$. With reparameterization, this can be formulated as $s_i[t] = H(u_i[t] - V_{th} - z_i)$ with a random noise variable $z_i$ that follows the distribution specified by $F$. Different $F$ corresponds to different distributions and noises. For example, the sigmoid function corresponds to a logistic noise, while the erf function corresponds to a Gaussian noise. Under the stochastic setting, the local surrogate derivatives can be introduced for the spiking function (Shekhovtsov & Yanush, 2021; Ma et al., 2023).

Specifically, consider the objective function which should turn to the expectation over random variables under the stochastic model. Considering a one-hidden-layer network with one time step, with the input $\mathbf{x}$ connecting to $n$ spiking neurons by the weight $\mathbf{W}$ and the neurons connecting to an output readout layer by the weight $\mathbf{O}$. Different from deterministic models with the objective function $\mathbb{E}_{\mathbf{x}}[\mathcal{L}(\mathbf{s})]$, where $\mathbf{s} = H(\mathbf{u} - V_{th}), \mathbf{u} = \mathbf{W}\mathbf{x}$, under the stochastic setting, the objective is to minimize:

$$\mathbb{E}_{\mathbf{x}}[\mathbb{E}_{\mathbf{s}\sim p(\mathbf{s}|\mathbf{x},\mathbf{W})}[\mathcal{L}(\mathbf{s})]]. \tag{16}$$

For this objective, the model can be differentiable and gradients can be derived (Shekhovtsov & Yanush, 2021; Ma et al., 2023). We focus on the gradients of $\mathbf{u}$, which can be expressed as:

$$\begin{aligned}
\frac{\partial}{\partial \mathbf{u}} \mathbb{E}_{\mathbf{s}\sim p(\mathbf{s}|\mathbf{W})}[\mathcal{L}(\mathbf{s})] &= \frac{\partial}{\partial \mathbf{u}} \sum_{\mathbf{s}} \left( \prod_i p(\mathbf{s}_i|\mathbf{W}) \right) \mathcal{L}(\mathbf{s}) \\
&= \sum_{\mathbf{s}} \sum_i \left( \prod_{i'\neq i} p(\mathbf{s}_{i'}|\mathbf{W}) \right) \left( \frac{\partial}{\partial \mathbf{u}} p(\mathbf{s}_i|\mathbf{W}) \right) \mathcal{L}(\mathbf{s}).
\end{aligned} \tag{17}$$

Then consider derandomization to perform summation over $s_i$ while keeping other random variables fixed (Shekhovtsov & Yanush, 2021). Let $\mathbf{s}_{\neg i}$ denote other variables except $s_i$. Since $s_i$ is $\{0, 1\}$ valued, given $\mathbf{s}_{\neg i}$, we have

$$\begin{aligned}
\sum_{s_i \in \{0,1\}} \frac{\partial p(s_i|\mathbf{W})}{\partial \mathbf{u}} \mathcal{L}([\mathbf{s}_{\neg i}, s_i]) &= \frac{\partial p(\mathbf{s}_i|\mathbf{W})}{\partial \mathbf{u}} \mathcal{L}(\mathbf{s}) + \frac{\partial (1 - p(\mathbf{s}_i|\mathbf{W}))}{\partial \mathbf{u}} \mathcal{L}(\mathbf{s}_{\downarrow i}) \\
&= \frac{\partial p(\mathbf{s}_i|\mathbf{W})}{\partial \mathbf{u}} \left( \mathcal{L}(\mathbf{s}) - \mathcal{L}(\mathbf{s}_{\downarrow i}) \right),
\end{aligned} \tag{18}$$

where $\mathbf{s}$ is a random sample considering $s_i$ (the RHS is invariant of $s_i$), and $\mathbf{s}_{\downarrow i}$ denotes taking $\mathbf{s}_i$ as the other state for $\mathbf{s}$. Given that $\sum_{s_i} p(s_i|\mathbf{W}) = 1$, Eq. (17) is equivalent to

$$\begin{aligned}
\frac{\partial}{\partial \mathbf{u}} \mathbb{E}_{\mathbf{s}\sim p(\mathbf{s}|\mathbf{W})}[\mathcal{L}(\mathbf{s})] &= \sum_i \sum_{\mathbf{s}_{\neg i}} \left( \prod_{i'\neq i} p(\mathbf{s}_{i'}|\mathbf{W}) \right) \sum_{s_i} \left( \frac{\partial}{\partial \mathbf{u}} p(s_i|\mathbf{W}) \right) \mathcal{L}([\mathbf{s}_{\neg i}, s_i]) \\
&= \sum_i \sum_{\mathbf{s}_{\neg i}} \left( \prod_{i'\neq i} p(\mathbf{s}_{i'}|\mathbf{W}) \right) \sum_{s_i} p(s_i|\mathbf{W}) \frac{\partial p(\mathbf{s}_i|\mathbf{W})}{\partial \mathbf{u}} \left( \mathcal{L}(\mathbf{s}) - \mathcal{L}(\mathbf{s}_{\downarrow i}) \right) \\
&= \sum_{\mathbf{s}} \left( \prod_i p(\mathbf{s}_i|\mathbf{W}) \right) \sum_i \frac{\partial p(\mathbf{s}_i|\mathbf{W})}{\partial \mathbf{u}} \left( \mathcal{L}(\mathbf{s}) - \mathcal{L}(\mathbf{s}_{\downarrow i}) \right) \\
&= \mathbb{E}_{\mathbf{s}\sim p(\mathbf{s}|\mathbf{W})} \sum_i \frac{\partial p(\mathbf{s}_i|\mathbf{W})}{\partial \mathbf{u}} \left( \mathcal{L}(\mathbf{s}) - \mathcal{L}(\mathbf{s}_{\downarrow i}) \right).
\end{aligned} \tag{19}$$

Taking one sample of $\mathbf{s}$ in each forward procedure allows the unbiased gradient estimation as the Monte Carlo method. In this equation, considering the probability distribution, we have:

$$\frac{\partial p(\mathbf{s}_i|\mathbf{W})}{\partial \mathbf{u}} = F'(\mathbf{u}, V_{th}), \tag{20}$$

where $F'$ is the derivative of $F$, corresponding to a *local* surrogate gradient, e.g., the derivative of the sigmoid function, triangular function, etc.

The term $\mathcal{L}(\mathbf{s}) - \mathcal{L}(\mathbf{s}_{\downarrow i})$ corresponds to the error, and the above derivation is also similar to REINFORCE (Williams, 1992). However, since it relies on derandomization, simultaneous perturbation is infeasible in this formulation, and for efficient simultaneous calculation of all components, we may follow previous works (Shekhovtsov & Yanush, 2021) to tackle it by linear approximation: $\mathcal{L}(\mathbf{s}) - \mathcal{L}(\mathbf{s}_{\downarrow i}) \approx \frac{\partial \mathcal{L}(\mathbf{s})}{\partial \mathbf{s}_i}$, enabling simultaneous calculation given a gradient $\frac{\partial \mathcal{L}(\mathbf{s})}{\partial \mathbf{s}}$. This approximation may introduce bias, while it can be small for over-parameterized neural networks with weights at the scale of $\frac{1}{\sqrt{d_n}}$, where $d_n$ is the neuron number. This means that for the elements of the readout $\mathbf{o} = \mathbf{O}\mathbf{s}$, flipping the state of $\mathbf{s}_i$ only has $O(\frac{1}{\sqrt{d_n}})$ influence.

The deterministic model may be viewed as a special case, e.g., with noise always as zero, and Shekhovtsov & Yanush (2021) show that the gradients under the deterministic setting can provide a similar ascent direction under certain conditions. Also, the noise injection in our method is similar to introducing the randomness in stochastic neuron model.

Therefore, spiking neurons can be differentiable under the stochastic setting and *local* surrogate derivatives can be well-defined, supporting our formulation as introduced in the main text. Our pseudo-zeroth-order method approximates $\frac{\partial \mathcal{L}(\mathbf{s})}{\partial \mathbf{s}}$, fitting the above formulation. Also note that the above derivation of surrogate derivatives is *local* for one hidden layer – for multi-layer networks, while we may iteratively perform the above analysis to obtain the commonly used global surrogate gradients, there can be expanding errors through layer-by-layer propagation due to the linear approximation error. Differently, our OPZO performs direct error feedback, which may reduce such errors.

## C  More Implementation Details

### C.1  Local Learning

For experiments with local learning, we consider local supervision with a fully connected readout for each layer. Specifically, for the output $\mathbf{s}^l$ of each layer, we calculate the local loss based on the readout $\mathbf{r}^l = \mathbf{R}^l \mathbf{s}^l$ as $\mathcal{L}(\mathbf{r}^l, \mathbf{y})$. Then the gradient for $\mathbf{s}^l$ is calculated by the local loss and added to the global gradient based on our OPZO method, which will update synaptic weights directly connected to the neurons. We assume the weight symmetry of local learning for propagating errors, i.e., using a feedback weight $\mathbf{P}^l = \mathbf{R}^l$ to propagate errors as $\mathbf{P}^{l\top} \frac{\partial \mathcal{L}(\mathbf{r}^l, \mathbf{y})}{\partial \mathbf{r}^l}$. This is because for the single linear layer that directly connects to the output, the weight $\mathbf{P}^l$ can be learned to be symmetric to $\mathbf{R}^l$ through symmetric local Hebbian-like update rule, i.e., both of them are updated by $\frac{\partial \mathcal{L}(\mathbf{r}^l, \mathbf{y})}{\partial \mathbf{r}^l} \mathbf{s}^{l\top}$ based on pre- and post-synaptic information (e.g., $\frac{\partial \mathcal{L}(\mathbf{r}^l, \mathbf{y})}{\partial \mathbf{r}^l} = \mathbf{r}^l - \mathbf{y}$ for MSE loss and $\frac{\partial \mathcal{L}(\mathbf{r}^l, \mathbf{y})}{\partial \mathbf{r}^l} = \sigma(\mathbf{r}^l) - \mathbf{y}$ for CE loss). This mechanism does not require global error information and is compatible with the intended constraints, while it is only applicable to the single (linear) layer condition. Kaiser et al. (2020) also show that a fixed random matrix can be effective for such kind of local learning.

We also consider intermediate global learning (IGL) as a kind of local learning. That is, we choose a middle layer to perform readout for loss calculation, just as the last layer, and its direct feedback signal will be propagated to previous layers. For the experiments with a 9-layer network, we choose the middle layer as the fourth convolutional layer.

### C.2  Noise Injection

For each time step of SNNs, we sample a $z$ and add it to the network after or before the neural activities (see Section 4.4). Compared with the two-point zeroth-order estimation, the considered one-point method can have a much larger variance. To further reduce the variance, we can leverage antithetic $\mathbf{z}$, i.e., $\mathbf{z}$ and $-\mathbf{z}$, for every two time steps of SNNs. Since SNNs naturally have multiple

time steps and the inputs for different time steps usually belong to the same object with similar distributions, this approach may roughly approximate the two-point formulation without additional costs.

### C.3 TRAINING SETTINGS

#### C.3.1 DATASETS

We conduct experiments on N-MNIST (Orchard et al., 2015), DVS-Gesture (Amir et al., 2017), DVS-CIFAR10 (Li et al., 2017), MNIST (LeCun et al., 1998), CIFAR-10 and CIFAR-100 (Krizhevsky & Hinton, 2009), as well as ImageNet (Deng et al., 2009).

**N-MNIST** N-MNIST is a neuromorphic dataset converted from MNIST by a Dynamic Version Sensor (DVS), with the same number of training and testing samples as MNIST. Each sample consists of spike trains triggered by the intensity change of pixels when DVS scans a static MNIST image. There are two channels corresponding to ON- and OFF-event spikes, and the pixel dimension is expanded to $34 \times 34$ due to the relative shift of images. Therefore, the size of the spike trains for each sample is $34 \times 34 \times 2 \times T$, where $T$ is the temporal length. The original data record $300ms$ with the resolution of $1\mu s$. We follow Zhang & Li (2020) to reduce the time resolution by accumulating the spike train within every $3ms$ and use the first 30 time steps. The license of N-MNIST is the Creative Commons Attribution-ShareAlike 4.0 license.

**DVS-Gesture** DVS-Gesture is a neuromorphic dataset recording 11 classes of hand gestures by a DVS camera. It consists of 1,176 training samples and 288 testing samples. Following Fang et al. (2021), we pre-possess the data to integrate event data into 20 frames, and we reduce the spatial resolution to $48 \times 48$ by interpolation. The license of DVS-Gesture is the Creative Commons Attribution 4.0 license.

**DVS-CIFAR10** DVS-CIFAR10 is the neuromorphic dataset converted from CIFAR-10 by DVS, which is composed of 10,000 samples, one-sixth of the original CIFAR-10. It consists of spike trains with two channels corresponding to ON- and OFF-event spikes. We split the dataset into 9000 training samples and 1000 testing samples as the common practice, and we reduce the temporal resolution by accumulating the spike events (Fang et al., 2021) into 10 time steps as well as the spatial resolution into $48 \times 48$ by interpolation. We apply the random cropping augmentation similar to CIFAR-10 to the input data and normalize the inputs based on the global mean and standard deviation of all time steps. The license of DVS-CIFAR10 is CC BY 4.0.

**MNIST** MNIST consists of 10-class handwritten digits with 60,000 training samples and 10,000 testing samples. Each sample is a $28 \times 28$ grayscale image. We normalize the inputs based on the global mean and standard deviation, and convert the pixel value into a real-valued input current at every time step. The license of MNIST is the MIT License.

**CIFAR-10** CIFAR-10 consists of 10-class color images of objects with 50,000 training samples and 10,000 testing samples. Each sample is a $32 \times 32 \times 3$ color image. We normalize the inputs based on the global mean and standard deviation, and apply random cropping, horizontal flipping, and cutout (DeVries & Taylor, 2017) for data augmentation. The inputs to the first layer of SNNs at each time step are directly the pixel values, which can be viewed as a real-valued input current.

**CIFAR-100** CIFAR-100 is a dataset similar to CIFAR-10 except that there are 100 classes of objects. It also consists of 50,000 training samples and 10,000 testing samples. We use the same pre-processing as CIFAR-10.

The license of CIFAR-10 and CIFAR-100 is the MIT License.

**ImageNet** ImageNet-1K is a dataset of color images with 1,000 classes of objects, containing 1,281,167 training samples and 50,000 validation images. We adopt the common pre-possessing strategies to first randomly resize and crop the input image to $224 \times 224$, and then normalize it after the random horizontal flipping data augmentation, while the testing images are first resized to

$256 \times 256$ and center-cropped to $224 \times 224$, and then normalized. The inputs are also converted to a real-valued input current at each time step. The license of ImageNet is Custom (non-commercial).

### C.3.2 TRAINING DETAILS AND HYPERPARAMETERS

For SNN models, following the common practice, we leverage the accumulated membrane potential of the neurons at the last classification layer (which will not spike or reset) for classification, i.e., the classification during inference is based on the accumulated $\mathbf{u}^N[T] = \sum_{t=1}^{T} \mathbf{o}[t]$, where $\mathbf{o}[t] = \mathbf{W}^{N-1}\mathbf{s}^{N-1}[t] + \mathbf{b}^N$ which can be viewed as an output at each time step. The loss during training is calculated for each time step as $\mathcal{L}(\mathbf{o}[t], \mathbf{y})$ following the instantaneous loss in online training with the loss function as a combination of cross-entropy (CE) loss and mean-square-error (MSE) loss (Xiao et al., 2022). For spiking neurons, $V_{th} = 1$ and $\lambda = 0.5$. We leverage the sigmoid-like local surrogate derivative, i.e., $\psi(u) = \frac{1}{a_1} \frac{e^{(V_{th}-u)/a_1}}{(1+e^{(V_{th}-u)/a_1})^2}$ with $a_1 = 0.25$. For convolutional networks, we apply the scaled weight standardization (Brock et al., 2021) as in Xiao et al. (2022).

For our OPZO method, as well as the ZO method in experiments, $\alpha$ is by default set as $0.2$ initially and linearly decays to $0.01$ through the epochs, in order to reduce the influence of stochasticness for forward propagation. In practice, this schedule is not critical (see analysis in Section D.4). For fine-tuning on ImageNet under noise, $\alpha$ is set as the noise scale, and we do not apply antithetic variables across time steps, in order to better fit the noisy test setting (perturbation noise is before the neuron). In practice, we remove the factor $1/\alpha$ for the calculation of $\mathbf{M}$, because in the single-point setting, the scale of $\tilde{\mathbf{o}}$ is larger than and not proportional to $\alpha$. This only influences the estimated gradient with a scale $\alpha$, and may be offset by the adaptive optimizer. Analysis experiments show that this factor is also not critical (see Section D.4). For gradient variance analysis, we keep this factor in order to have a comparable gradient scale.

For N-MNIST and MNIST, we consider FC networks with two hidden layers composed of 800 neurons, and for DVS-CIFAR10, DVS-Gesture, CIFAR-10, and CIFAR-100, we consider 5-layer Conv networks (128C3-AP2-256C3-AP2-512C3-AP2-512C3-FC), or 9-layer Conv networks under the deeper network setting (64C3-128C3-AP2-256C3-256C3-AP2-512C3-512C3-AP2-512C3-512C3-FC). We train our models on common datasets by the AdamW optimizer with learning rate 2e-4 and weight decay 2e-4 (except for ZO, the learning rate is set as 2e-5 on DVS-CIFAR10, MNIST, CIFAR-10, and CIFAR-100 for better results). The batch size is set as 128 for most datasets and 16 for DVS-Gesture, and the learning rate is cosine annealing to 0. For N-MNIST and MNIST, we train models by 50 epochs and we apply dropout with the rate 0.2 (except for ZO). For DVS-Gesture, DVS-CIFAR10, CIFAR-10, and CIFAR-100, we train models by 300 epochs. For DVS-CIFAR10, we apply dropout with the rate 0.1 (except for ZO). We set the momentum coefficient for momentum feedback connections as $\lambda = 0.99999$ (except for DVS-Gesture, it is set as $\lambda = 0.999999$ due to a smaller batch size), and for the combination with local learning, the local loss is scaled by $0.01$.

For fine-tuning ImageNet, the learning rate is set as 2e-6 (and 2e-7 for ZO) without weight decay, and the batch size is set as 64. The perturbation noise is before the neuron, i.e., added to the results after convolutional operations. For BP, we train 1 epoch. For DFA, ZO, and OPZO, we train 5 epochs. We observe that DFA and ZO fail after 1 epoch, so we only report the results after 1 epoch, and for OPZO, the results can continually improve, so we report the results after 5 epochs. The 1-epoch and 5-epoch results for OPZO are 63.04 and 63.39 under the noise scale of 0.1, and 59.50 and 60.96 under the noise scale of 0.15.

The code implementation is based on the PyTorch framework, and experiments are carried out on one NVIDIA GeForce RTX 3090 GPU (each experiment takes several hours). Experiments are based on 3 runs of experiments with the same random seeds 2022, 0, and 1. Note that the results hardly change for more runs of experiments: for OPZO results on DVS-CIFAR10 with the largest standard deviation, 10 runs have almost the same results ($72.69 \pm 0.62$ vs. $72.77 \pm 0.82$).

For gradient variance experiments, the variances are calculated by the batch gradients in one epoch, i.e., $var = \frac{\sum \|\mathbf{g}_i - \bar{\mathbf{g}}\|^2}{n}$, where $\mathbf{g}_i$ is the batch gradient, $\bar{\mathbf{g}}$ is the average of batch gradients, and $n$ is the number of batches multiplied by the number of elements in the gradient vector.

**DFA and DKP** For DFA, the direct error feedback weight is randomly initialized following the Kaiming initialization strategy. DKP (Webster et al., 2020) is based on the formulation of DFA

Table 6: Brief comparison of training costs on GPU for CIFAR-10 with convolutional networks. $^\dagger$ means manual implementation of spatial BP with layer-by-layer backpropagation, which is in a similar fashion as other methods. $^\ddagger$ means using automatic differentiation implemented by PyTorch with low-level code optimizations.

| Method | Memory | Time per epoch |
|---|---|---|
| Spatial BP | $2.8G^\dagger$ / $2.9G^\ddagger$ | $49s^\dagger$ / $45s^\ddagger$ |
| DFA | 2.8G | 44s |
| $ZO_{sp}$ | 2.8G | 46s |
| OPZO | 2.9G | 46s |
| DFA (w/ LL) | 3.0G | 50s |
| OPZO (w/ LL) | 3.1G | 51s |

and updates feedback weights similar to Kolen-Pollack learning, which calculates gradients for feedback weights by the product of the middle layer's activation and the error from the top layer. The feedback weights are initialized as zero, and we treat them as parameters to be optimized by the Adam optimizer. Its basic thought is trying to keep the update direction of feedback and feedforward weights the same, but it may lack sufficient theoretical grounding. As DKP is designed for ANN, we implement it for SNN with the adaptation of activations to pre-synaptic traces for feedback weight learning (similar to the update of feedforward weight). As shown in the results, compared with DFA, DKP can have around 2-3% performance improvement on CIFAR-10 and CIFAR-100, which is similar to the improvement in its paper. However, DKP cannot work well for neuromorphic datasets. And OPZO significantly outperforms both DKP and DFA on all datasets.

# D    ADDITIONAL RESULTS

## D.1    TRAINING COSTS ON GPU

We provide a brief comparison of memory and time costs of different methods on GPU in Table 6. Our proposed OPZO has about the same costs as spatial BP and DFA. If we exclude some code-level optimization and implement all methods in a similar fashion, DFA and OPZO are faster than spatial BP, which is consistent with the theoretical analysis of operation numbers. Note that this is only a brief comparison, as we do not perform low-level code optimization for OPZO and DFA, for example, the direct feedback of OPZO and DFA to different layers can be parallel, and local learning for different layers can also theoretically be parallel, to further reduce the time. As described in the main text, the target of neuromorphic computing with SNNs would be potential neuromorphic hardware, and OPZO and DFA can have lower costs, while GPUs generally do not follow the properties. Since neuromorphic hardware is still under development and we have limited access, we mainly simulate the experiments on GPUs, and it can be future work to consider the combination with neuromorphic hardware implementation.

Also please note that these methods are all based on online training, so the memory costs (agnostic to time steps) are already largely reduced compared with BPTT (proportional to time steps) (Xiao et al., 2022).

## D.2    FIRING RATE AND SYNAPTIC OPERATIONS

For event-driven SNNs, the energy costs on neuromorphic hardware are proportional to the spike count, or more precisely, synaptic operations induced by spikes. Therefore, we also compare the firing rate (i.e., average spike count per neuron per time step) and synaptic operations of the models trained by different methods. As shown in Table 7, on both DVS-CIFAR10 and CIFAR-10, OPZO (w/ LL) achieves the lowest average total firing rate and synaptic operations, indicating the most energy efficiency. The results also demonstrate different spike patterns for models trained by different methods, and show that LL can significantly improve OPZO, while it can hardly improve DFA and spatial BP. It may indicate OPZO as a better, more biologically plausible global learning method to be combined with local learning.

Table 7: The firing rate (fr) and synaptic operations (SynOp) induced by spikes for models trained by different methods on DVS-CIFAR10 and CIFAR-10.

| DVS-CIFAR10 | | | | | | |
|---|---|---|---|---|---|---|
| Method | Layer1 fr | Layer2 fr | Layer3 fr | Layer4 fr | Total fr | SynOp |
| Spatial BP | 0.1763 | 0.1733 | 0.2394 | 0.3575 | 0.1904 | $1.42 \times 10^9$ |
| Spatial BP (w/ LL) | 0.2272 | 0.1618 | 0.2199 | 0.3439 | 0.2122 | $1.51 \times 10^9$ |
| DFA | 0.2693 | 0.4564 | 0.4783 | 0.4930 | 0.3574 | $2.91 \times 10^9$ |
| DFA (w/ LL) | 0.2433 | 0.4531 | 0.4848 | 0.4919 | 0.3430 | $2.84 \times 10^9$ |
| OPZO | 0.2435 | 0.3446 | 0.4212 | 0.4222 | 0.3021 | $2.41 \times 10^9$ |
| OPZO (w/ LL) | 0.0406 | 0.0614 | 0.1451 | 0.2838 | **0.0691** | $\mathbf{0.53 \times 10^9}$ |
| CIFAR-10 | | | | | | |
| Method | Layer1 fr | Layer2 fr | Layer3 fr | Layer4 fr | Total fr | SynOp |
| Spatial BP | 0.2005 | 0.1679 | 0.1067 | 0.0493 | 0.1734 | $0.76 \times 10^9$ |
| Spatial BP (w/ LL) | 0.1870 | 0.1474 | 0.0978 | 0.0470 | 0.1589 | $0.69 \times 10^9$ |
| DFA | 0.1769 | 0.3787 | 0.4314 | 0.4149 | 0.2759 | $1.40 \times 10^9$ |
| DFA (w/ LL) | 0.1196 | 0.3180 | 0.4089 | 0.3878 | 0.2235 | $1.16 \times 10^9$ |
| OPZO | 0.1563 | 0.2861 | 0.3496 | 0.2754 | 0.2229 | $1.12 \times 10^9$ |
| OPZO (w/ LL) | 0.0400 | 0.0670 | 0.1159 | 0.2157 | **0.0640** | $\mathbf{0.30 \times 10^9}$ |

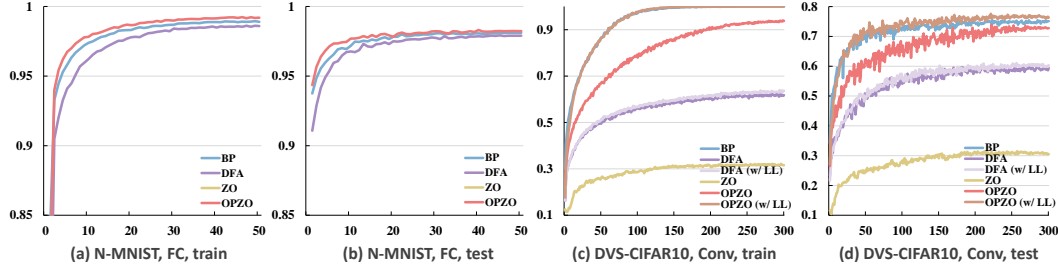

Figure 3: Training dynamics of different methods on N-MNIST and DVS-CIFAR10.

### D.3 TRAINING DYNAMICS AND GRADIENT SIMILARITY

We present the training dynamics of different methods in Fig. 3. For the fully connected network on N-MNIST, OPZO achieves a similar convergence speed as spatial BP, which is better than DFA and much better than ZO. For the convolutional network on DVS-CIFAR10, OPZO itself is slower than spatial BP while still performing much better than DFA and ZO, and when combined with local learning, OPZO (w/ LL) achieves a similar training convergence speed as spatial BP as well as a better testing performance.

We further present the cosine similarity between estimated gradients and backpropagated gradients with surrogate derivatives in Fig. 4. The results show that the cosine similarity of different layers between OPZO and BP remains in the range of 0.5-0.9 throughout training, whereas DFA and BP is typically below 0.1 for most layers. This indicates that the bias introduced by momentum feedback does not significantly distort the gradient direction compared to DFA, and the training can converge with effective descent directions.

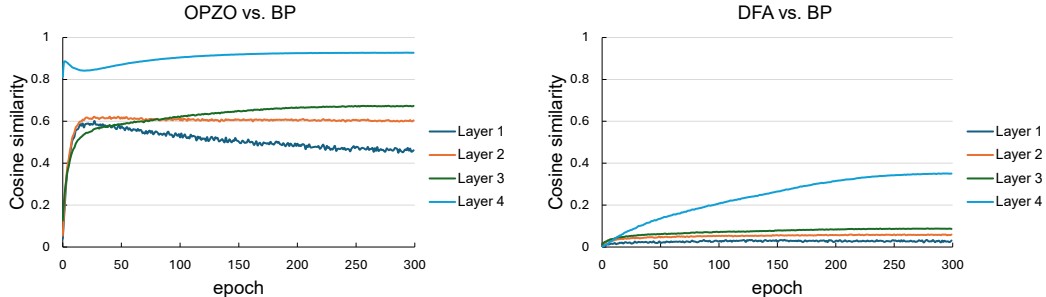

Figure 4: Cosine similarity between OPZO and BP with surrogate derivatives as well as between DFA and BP on CIFAR-100.

Table 8: Analysis results of different $\lambda$ on CIFAR-100.

| $\lambda = 0$ (w/o momentum) | $\lambda = 0.9$ | $\lambda = 0.99$ | $\lambda = 0.999$ | $\lambda = 0.9999$ | $\lambda = 0.99999$ |
|---|---|---|---|---|---|
| 16.08±0.37 | 39.05±0.69 | 50.65±0.05 | 58.25±0.33 | 60.92±0.11 | 60.93±0.16 |

Table 9: Analysis results of updating $\mathbf{M}$ throughout training on CIFAR-100.

| stop updating $\mathbf{M}$ after 10 epochs | OPZO |
|---|---|
| 57.17 | 60.93±0.16 |

Table 10: Analysis results of settings with initial perturbation scale $\alpha = 0.2$ on CIFAR-100.

| w/ schedule | w/o schedule | w/o schedule and w/ $\frac{1}{\alpha}$ factor |
|---|---|---|
| 60.93±0.16 | 61.85±0.09 | 61.65±0.40 |

Table 11: Analysis results of different perturbation scale $\alpha$ without scheduling on CIFAR-100.

| $\alpha = 1.$ | $\alpha = 0.2$ | $\alpha = 0.02$ | $\alpha = 0.002$ |
|---|---|---|---|
| 58.34 | 61.85±0.09 | 58.33 | 49.97 |

## D.4 ANALYSIS OF HYPERPARAMETERS

We first study the influence of the momentum coefficient $\lambda$ in Table 8. As shown in the results, OPZO requires a large $\lambda$ in our scenario. This is due to the large variance of single-point zeroth-order approximation, so we require a relatively small $1-\lambda$ for smoothing $\mathbf{M}$ to approximate the expectation $\mathbb{E}_{\mathbf{x}} \left[ \mathbf{J}_f^\top(\mathbf{x}) \right]$ (more precisely, $\mathbb{E}_{\mathbf{x}} \left[ \mathbf{J}_{f_\alpha}^\top(\mathbf{x}) \right]$). A smaller $\lambda$ cannot properly deal with the large variance, leading to inferior performance. While $\lambda$ is large, this does not mean $\mathbf{M}$ is quasi-static, because the objective of the expectation of Jacobian is slowly changing throughout training. To validate this in experiments, we stop updating $\mathbf{M}$ after 10 epochs, and the performance drops as shown in Table 9.

We then analyze the influence of the perturbation scale $\alpha$. We first evaluate the effect of the scheduling of $\alpha$ and removing the factor $\frac{1}{\alpha}$ (Section C.3.2). As shown in Table 10, the scheduling has slightly negative influence on the performance while the $\frac{1}{\alpha}$ factor has negligible impact. We further analyze different perturbation scales without scheduling in Table 11. As shown in the results, the scale around 0.2 works best. This is likely due to the property of spiking neural networks, where we set the spiking threshold as 1: if the perturbation scale is too small, the perturbation hardly influences the spiking generation, leading to imprecise estimation. Therefore, for spiking neural networks, a good scale choice would be fixed $\alpha = 0.2$ throughout the training.

Table 12: Analysis results of deeper networks without auxilary techniques on CIFAR-100.

| Network structure | Spatial BP | DFA | $ZO_{sp}$ | OPZO |
|---|---|---|---|---|
| 5-layer network | 64.82±0.09 | 49.50±0.13 | 22.26±0.51 | 60.93±0.16 |
| 9-layer network | 65.96±0.52 | 48.14±0.49 | 4.85±0.21 | 56.89±0.37 |
| 9-layer network (w/ residual) | 69.5 | 48.5 | / | 63.4 |
| 18-layer residual network (smaller channels) | 66.78 | 46.83 | / | 61.86 |
| 34-layer residual network (smaller channels) | 67.36 | 45.06 | / | 61.17 |

### D.5 Analysis of Pure OPZO for Training Deeper Networks from Scratch

In this section, we provide more analysis on the pure OPZO for training deeper networks from scratch without auxiliary techniques, as shown in Table 12.

First, for the plain 9-layer network, OPZO has degraded performance compared with 5-layer network, but still significantly outperforming DFA and ZO. This is related to our theoretical analysis of the bias of average Jacobian, which shows that the smoothness will influence the condition of effective descent direction (Remark A.5). The plain deeper networks' unsmoothness tightens the condition, leading to inferior performance. So some auxiliary mechansims may be required to alleviate the problem.

Then, we further show that residual connections can largely alleviate the problem, in consistent with our theoritical analysis and experiments that pure OPZO can effectively fine-tune ResNet-34 on ImageNet. As shown in Table 12, with residual connections, OPZO does not degrade as depth grows and can scale to 34-layer networks in the train from scratch setting as well, while DFA still has degraded performance. This is because residual connections can make the network function smoother, so the problem is largely alleviated. While there is still some performance gap with BP, OPZO significantly outperforms DFA and can be further enhanced with auxiliary techniques.

Therefore, pure OPZO also has the ability to train deeper networks from scratch, while it has more reliance on the proper network structure (residual connections) than BP.

Additionally, we report results for training a NF-ResNet18 model (T=4) from scracth on ImageNet for 100 epochs using the Adam optimizer (lr=$1e-4$). The performance of pure OPZO, DFA, and BP is 19.0%, 5.6%, and 52.9%, respectively. The results show that while OPZO significantly outperforms DFA, it still has much room for improvement compared with BP in this large-scale offline training, likely due to a more non-smooth optimization landscape as indicated by our theoretical analysis. This suggests that OPZO may require additional techniques such as local learning for this large-scale training-from-scratch scenario. However, we emphasize that OPZO is not intended to replace BP for high-compute offline training, which is orthogonal to our neuromorphic objective. Our goal is hardware-friendly, on-chip learning for neuromorphic SNNs, where starting from scratch is rarely necessary (similar to our brains that adapt rather than relearn entirely), and post-deployment adaptation and continual learning are key scenarios. Our ImageNet fine-tuning experiments show that pure OPZO can scale to large networks without auxiliary techniques, validating its effectiveness in this setting. This demonstrates OPZO's complementary role to offline BP methods.

## E More Discussions

### E.1 Limitations

This paper mainly focuses on theoretical groundings and simulation experiments with GPUs for the proposed method, while no implementation on neuromorphic hardware is included due to our limited access to it. Future work can consider the implementation on those hardware with more engineering efforts, e.g., on Loihi2 (Davies, 2021) that can support three-factor learning rules as described in their technical report.

As discussed in Section 5, our method is a different line from many recent SNN works with state-of-the-art performance, focusing on more biologically plausible and hardware-friendly training

Table 13: Results of different methods without truncating temporal gradients on DVS-CIFAR10.

| Network structure | STBP | DFA | PZO |
|---|---|---|---|
| 5-layer CNN (sWS) | 76.17±0.21 | 60.00±0.22 | 73.20±0.08 |
| 5-layer CNN (BN) | 77.23±0.46 | 60.77±0.69 | 74.13±0.52 |

algorithms. So we mainly evaluate the effectiveness of the method with comparisons under various settings, not pursuing the state-of-the-art performance. On the other hand, as discussed in the experiments of fine-tuning ResNet-34 on ImageNet, our method may be combined with those works aiming at state-of-the-art performance through the potential on-chip fine-tuning after deployment.

### E.2 Discussion of the Method

The proposed OPZO is built on online training methods to deal with the spatio-temporal locality problem of BP(TT) for neuromorphic computing and pave the path to on-chip SNN training. While the pseudo-zeroth-order formulation can be applied to non-online scenarios, e.g., BPTT, we do not focus on it because this is not our target (friendly for neuromorphic hardware and more biologically plausible) and it requires larger memory costs to maintain intermediate states through time for direct error propagation. Nevertheless, to further validate the feasibility of applying PZO to the setting without truncating the temporal gradient flow, we provide more results under this setting. Specifically, we leverage PZO to estimate gradients from the network output at time step $t_i$ to intermediate layers at time steps $t_j$ ($t_j \leq t_i$). To reduce the momentum costs, we share the momentum for the same interval between $t_i$ and $t_j$, i.e., there will be $T$ feedback momentum representing the feedback matrix from $t_i$-output to ($t_i - k$)-features ($k = 0, 1, \cdots, T-1$). We perform noise injection to all time steps and update the momentum feedback matrices based on noises and network outputs. Without the requirement for memory-efficient online training, we can also adopt BN (along all time steps) for networks. As shown in Table 13, PZO can be effectively applied to this setting.

We consider node perturbation instead of weight perturbation because it improves the latter with smaller variance (Lillicrap et al., 2020) and is more biologically plausible with a better analog to three-factor Hebbian learning (Frémaux & Gerstner, 2016) as discussed in Section 4.3. This is friendly for neuromorphic hardware that supports three-factor rules (Davies, 2021). While weight perturbation may be conceptually easier, it is less effective to optimize neural networks and has hardly been adopted in zeroth-order methods for neural networks (Jiang et al., 2024) except in specially designed fine-tuning settings (Malladi et al., 2023).

While this paper mainly considers SNNs, the proposed pseudo-zeroth-order formulation can also be applied to ANNs. If we consider the general computer, there can be techniques to reduce the memory overhead of the momentum feedback, such as low-rank approximation, only saving some output vectors and recomputing the matrix by re-drawing the perturbation with random seeds, etc. This paper mainly focuses on neuromorphic computing, and we leave the extension to more settings as future work.

### E.3 Relation to Neuroscientific Evidence

In this section, we discuss more on the neuroscientific evidence for noise injection and top-down signals in our method.

**Noise injection.** The biological systems are inherently noisy and it has long been recognized that noise can be utilized as a resource for computation and learning (Seung, 2003; Fiete & Seung, 2006; Maass, 2014; Lillicrap et al., 2020). Considering the perturbation with noise injection, it can be related to stochastic synaptic transmission (Seung, 2003) or "empiric" synapses carrying perturbing input from another part of the brain (Fiete & Seung, 2006). Such mechanisms provide the biological basis for perturbation learning, which is believed to be employed by the brain for some kinds of learning (Lillicrap et al., 2020). Our work builds on this zeroth-order perturbation, while introducing momentum feedback to solve the large variance problem of it.

**Top-down feedback.** In the three-factor Hebbian learning, synaptic updates are modulated by reward-prediction errors (RPE) and can be gated by feedback (FB) from higher brain regions through top-down feedback connections, leading to the update rule $\Delta w_{i,j} = \beta \cdot f_i(a_i) \cdot f_j(a_j) \cdot RPE \cdot FB_j$ (Roelfsema & Holtmaat, 2018). Anatomically, feedback projections originate from higher cortical areas and mostly provide input to superficial (L1–L3) and deep (L5) layers of lower sensory areas, targeting apical dendrites of pyramidal neurons and specific microcircuits (Roelfsema & Holtmaat, 2018). These feedback pathways are thought to play a key role in gating plasticity, credit assignment, and context-dependent modulation. The momentum feedback connections in our method are analogous to these top-down feedback pathways that modulate prediction errors, providing a basis for rules similar to three-factor Hebbian learning. Notably, the update of our feedback connections depends only on local pre- and post-synaptic, maintaining the simplicity and biological plausibility of the learning process.

