# OpenReview forum: "Online Pseudo-Zeroth-Order Training of Neuromorphic Spiking Neural Networks"
_ICLR.cc/2026/Conference — ICLR 2026 Poster_

### Official Review · Reviewer_Pjav · 2025-10-23

**Soundness:** 3
**Presentation:** 3
**Contribution:** 3
**Rating:** 6
**Confidence:** 3

**Summary:**

The paper proposes a method named OPZO for training Spiking Neural Networks (SNNs), which replaces spatial backpropagation with a single noised forward pass and momentum-updated feedback connections, aiming to achieve efficient and hardware-friendly credit assignment.

**Strengths:**

The OPZO method is novel and effective, skillfully addressing the high variance problem of traditional zeroth-order methods by retaining the first-order information of the loss function. Through extensive experiments, the paper demonstrates that OPZO's performance is significantly superior to other bio-plausible methods (like DFA) and can approach or even match the level of backpropagation (BP), offering a highly promising direction for efficient on-chip SNN training.

**Weaknesses:**

1.The provided ImageNet experiment is limited to fine-tuning a pre-trained network, which fails to demonstrate its performance on large-scale networks and datasets trained from scratch. For large-scale networks, OPZO might heavily rely on Local Learning to achieve performance comparable to BP.

2.Table 8 shows that OPZO requires an extremely large $\lambda$ for stable training. According to Equation 6, this implies that the feedback matrix M is updated very slowly. This raises the question: what would the performance be if a static M (i.e.,$\lambda=1$ after an initial phase) were used?

3.The paper lacks details on the initialization of the feedback matrix M, which weakens the method's reproducibility.

**Questions:**

1.OPZO and all compared methods are built upon the OTTT framework, which truncates the gradient flow across the time dimension. Is this a fundamental limitation for OPZO? Could OPZO be applied in a setting without truncating the temporal gradient flow (e.g., with full STBP)?

2.The paper mentions that the pseudo-zeroth-order method introduces bias but does not sufficiently analyze its impact on the final performance. Could you provide more theoretical analysis or experimental evidence on how this bias affects training stability and final accuracy?

3.Could you explain the rationale for removing the $1/\alpha$ factor in the practical implementation and discuss its impact on convergence?

---

> ### Author Response · Authors · 2025-11-20
> **Response to Reviewer Pjav**
>
> Thank you for appreciating our work and providing valuable comments. We respond to your comments and questions as follows.
>
> 1. ImageNet experiment trained from scratch.
>
> Thank you for your question. Actually, our work targets hardware-friendly, potential on-chip training for neuromorphic SNNs, so the ImageNet experiment was chosen to evaluate the scenario most relevant to this goal: post-deployment adaptation/training. Training from scratch on ImageNet with BP is an offline, high-compute regime that is orthogonal to our neuromorphic objective. We do not intend to replace BP&SG in offline training, but to provide a more suitable alternative for on-chip learning that is important for neuromorphic computing systems. This is related to applications such as mitigating hardware mismatch or online continual learning on hardware, which is orthogonal to offline training.
>
> In practice, it is not necessary for neuromorphic online (on-chip) learning to always start from scratch (similar to how our brain adapts rather than trained from scratch). Our experiments show that pure OPZO can effectively finetune large-scale networks without auxiliary techniques such as local learning (Table 4, note that Table 4 does not involve local learning), validating that OPZO can scale to large-scale networks in this setting. This demonstrates OPZO’s complementary role to offline BP methods.
>
> 2. Performance of static M.
>
> We have analyzed the static M in Appendix D.4 and Table 9. If we stop updating M after 10 epochs, the performance on CIFAR-100 drops from 60.93$\pm$0.16 to 57.17. This shows that while $\lambda$ is large, it does not mean M is static, because the objective of the expectation of Jacobian is slowly changing throughout training.
>
> 3. Details on the initialization of M and reproducibility.
>
> M is initialized as zero for our method and we have now specified it in Section 4.2. We have provided the full implementation code in supplementary materials for reproducibility.
>
> 4. Is OPZO limited to online training?
>
> No, the core idea of PZO is for spatial credit assignment and is not restricted to online training. We build the method on online training because our target is hardware-friendly algorithms for on-chip training, which has both temporal and spatial locality constraints.
>
> Here, we further supplement the results of applying PZO in the setting without truncating the temporal gradient flow. In this setting, we can also adopt BN (along all time steps) for networks which is restricted for online training. As shown in the results on DVS-CIFAR10 below, PZO can be effectively applied to this setting. We have added the results and details in Appendix E.2.
>
> | Network | Online-BP | Online-OPZO | Online-DFA | Full-BP | Full-PZO | Full-DFA |
> |:---:|:---:|:---:|:---:|:---:|:---:|:---:|
> | CNN (sWS) | 75.43$\pm$0.39 | 72.77$\pm$0.82 | 60.60$\pm$0.67 | 76.17$\pm$0.21 | 73.20$\pm$0.08 | 60.00$\pm$0.22 |
> | CNN (BN) | / | / | / | 77.23$\pm$0.46 | 74.13$\pm$0.52 | 60.77$\pm$0.69 |
>
> 5. The impact of bias on training stability and final performance.
>
> In our theoretical analysis, we provide results on conditions for descent direction (Proposition 4.3), which shows the influence of bias. Specifically, the bias introduces a dependency of effective descent direction on the smoothness of the network (Remark A.5), meaning that for highly non-smooth networks the descent direction remains effective only when the gradient norm is sufficiently large. This explains why pure OPZO relies more on proper network structures (such as residual connections) for scalability to deeper networks (analyzed in Appendix D.5). While the bias does not compromise training stability, it may lead to slightly lower final accuracy for highly non-smooth networks. For relatively smooth network, the bias does not significantly distort the gradient direction compared to other feedback methods.
>
> To support this, we have added empirical evidence in Appendix D.3: the cosine similarity of different layers between OPZO and BP (with surrogate gradients) remains in the range of 0.5-0.9 throughout training on CIFAR-100, whereas DFA and BP is typically below 0.1 for most layers. This indicates that OPZO maintains effective descent directions and the bias does not hinder training stability. While pure OPZO may show a slight performance drop due to the bias (Table 1), techniques such as local learning can mitigate this effect.
>
> 6. The 1/$\alpha$ factor.
>
> We remove this factor for the calculation of M because in the single-point zeroth-order setting, the scale of $\tilde{o}$ is larger than and not proportional to $\alpha$ (different from $\Delta o$ in the two-point setting which is expected to be proportional to $\alpha$). This only influences the estimated gradient with a scale $\alpha$, which will actually be offset by the adaptive optimizer and has negligible influence. We have analyzed this factor in Appendix D.4 and Table 10, and the results show that it has limited impact.

---

> > ### Comment · Reviewer_Pjav · 2025-11-25
> >
> > I appreciate your detailed rebuttal and the additional experiments. While I support positioning OPZO as a hardware-friendly on-chip training scheme, given the current reliance on GPU simulation, I still believe that reporting ImageNet training-from-scratch results would be valuable for a comprehensive comparison with other online methods.

---

> > > ### Author Response · Authors · 2025-12-02
> > >
> > > Thank you for your suggestion. We have included results and discussions in Appendix D.5.

---

### Official Review · Reviewer_EGrJ · 2025-10-28

**Soundness:** 2
**Presentation:** 3
**Contribution:** 3
**Rating:** 4
**Confidence:** 3

**Summary:**

This paper proposes online pseudo-zeroth-order (OPZO) training for spiking neural networks. Existing zero-order optimization methods have large variances. To address this issue, the authors introduced a pseudo-zero-order formulation and used momentum feedback connections. Moreover, to apply this method to SNNs, the authors replace the gradient estimation in the OTTT method with the proposed method. To demonstrate the effectiveness of their method, the authors conducted experiments on static and neuromorphic datasets.

**Strengths:**

1. The experimental results demonstrate that the proposed method effectively reduces variance.
2. Experimental results demonstrate that the proposed method exhibits robustness against different gradient noise injections.

**Weaknesses:**

1. The novelty of introducing a momentum mechanism in error propagation is limited.
2. It does not seem that the proposed method takes into account the unique properties of SNNs.
3. The performance of the proposed method is not consistently superior. The proposed method performs worse than OTTT [1], especially on static datasets such as CIFAR and ImageNet. The authors did not even compare the performance of the proposed method against OTTT on ImageNet. Furthermore, there is a lack of comparisons to other SOTA online learning or acceleration methods for SNNs.


```
[1] Online Training Through Time for Spiking Neural Networks. NeurIPS. 2022.
```

**Questions:**

See Weakness.

---

> ### Author Response · Authors · 2025-11-20
> **Response to Reviewer EGrJ**
>
> Thank you for your review and comments. We respond to your questions as follows.
>
> 1. Novelty of momentum mechanism in error propagation.
>
> To our knowledge, no previous work introduces momentum feedback in error propagation for zeroth-order methods. Existing momentum techniques mainly focus on accelerating gradient descent through momentum of gradients, but not how gradients are obtained through error propagation, especially for zeroth-order methods. Our work introduces a novel pseudo-zeroth-order formulation that decouples the model function and the loss function, thus enabling the use of momentum feedback connections to estimate the expectation of the Jacobian and propagate error signals. This mechanism is different from standard momentum in optimization.
>
> 2. It does not seem that the proposed method takes into account the unique properties of SNNs.
>
> We respectfully disagree with this point. The true value of SNNs (in AI) ultimately lies in the broader context of neuromorphic computing, where brain-inspired hardware-software codesign is crucial for building energy efficient systems, and SNNs should finally be combined with neuromorphic hardware for real efficiency. Therefore, the properties of neuromorphic computing, not only the event-driven spiking nature of neurons but also in-memory computation and local learning rules, are unique and critical properties for neuromorphic computing and SNNs. Our method is specifically designed for this neuromorphic paradigm, dealing with the spatial and temporal locality challenges of SNN learning on neuromorphic hardware and targeting on-chip training of SNNs. We take these unique properties into account, rather than focusing solely on the spiking nature.
>
> 3. The performance and comparisons.
>
> Our method is a different line from BP-based methods that pursue state-of-the-art (offline) performance. Instead, our target is to develop algorithms that are more hardware-friendly for potential on-chip training of neuromorphic SNNs. As discussed in Introduction, OPZO is designed to avoid some biologically implausible requirements of BP that are unfriendly for brain-like learning on neuromorphic hardware under locality constraints, while maintaining competitive performance. Our experiments are mainly intended to demonstrate the effectiveness and robustness of OPZO, rather than to outperform BP-based methods. Additionally, our experiments of fine-tuning ResNet-34 on ImageNet under noise further show the scenario where OPZO is orthogonal to and compatible with existing methods: we can first train high-performance models offline and then leverage OPZO for efficient on-chip adaptation, continual learning, or hardware mismatch correction. Our method is not to replace BP&SG in offline training, but provide a more suitable alternative for on-chip learning that is important for neuromorphic computing systems.
>
> Regarding OTTT, all our experiments use OTTT for temporal credit assignment (line 366-368; because this paper mainly focus on the spatial credit assignment problem), and we have compared OPZO to OTTT (i.e., the BP baseline in the results) in all relevant tables under the same settings, including ImageNet fine-tuning results in Table 4. Our method is an alternative to spatial BP and is compatible with other online temporal learning methods.
>
> Other SOTA online learning or acceleration methods for SNNs are still based on spatial BP and pursuing offline performance, not for hardware-friendly, on-chip learning. Our goal is to provide an alternative that is more suitable for neuromorphic hardware, and OPZO can be integrated with other online learning frameworks as well.

---

### Official Review · Reviewer_bmzH · 2025-10-31

**Soundness:** 3
**Presentation:** 3
**Contribution:** 3
**Rating:** 8
**Confidence:** 2

**Summary:**

This work proposes a pseudo-online zeroth-order training method that requires only a single forward pass, unlike existing approaches that rely on two forward passes. The authors achieve this by decoupling the loss function from the model and using the error signal (potentially a vector) computed at the final layer to update the model parameters. Additionally, a momentum feedback mechanism is introduced, which utilizes feedback signals from previous iterations to refine parameter updates. The authors claim that these modifications lead to improved performance and reduced variance in the training signal.

Experimentally, the proposed method demonstrates enhanced performance across multiple datasets and is compared against a range of baseline methods.

I am not an expert in zeroth-order optimization methods for SNN models, although I do have experience working with spiking neural networks. Therefore, I am assigning my score with low confidence.

**Strengths:**

The presentation of the paper is clear and well-organized.

This work addresses an important issue: the high variance problem in zeroth-order optimization methods.

The proposed decoupling approach is particularly interesting and provides a novel perspective on improving training stability and efficiency.

**Weaknesses:**

My main concern is that existing backpropagation or surrogate-gradient-based methods already achieve strong performance in similar settings. It is therefore not entirely clear why one should prefer the proposed zeroth-order approach over these established alternatives.

**Questions:**

Please see the weaknesses section.

---

> ### Author Response · Authors · 2025-11-20
> **Response to Reviewer bmzH**
>
> Thank you for appreciating our work and providing valuable comments. We respond to your questions as follows.
>
> 1. Position of the method.
>
> Our method is a different line from (spatiotemporal) BP-based methods that pursue state-of-the-art (offline) performance. Instead, our target is to develop algorithms that are more hardware-friendly for potential on-chip training of neuromorphic SNNs. As discussed in Introduction, OPZO is designed to avoid some biologically implausible requirements of BP that are unfriendly for brain-like learning on neuromorphic hardware under locality constraints, while maintaining competitive performance. Our experiments are mainly intended to demonstrate the effectiveness and robustness of OPZO, rather than to outperform BP-based methods. Notably, our experiments of fine-tuning ResNet-34 on ImageNet under noise further show the scenario where OPZO is orthogonal to and compatible with existing methods: we can first train high-performance models offline and then leverage OPZO for efficient on-chip adaptation, continual learning, or hardware mismatch correction. In summary, our method is not to replace BP&SG in offline training, but provide a more suitable alternative for on-chip learning that is important for neuromorphic computing systems.

---

> > ### Comment · Reviewer_bmzH · 2025-11-26
> > **Thanks for your response**
> >
> > Thanks for your response. I have already given a high score, and I will stick with that.

---

### Official Review · Reviewer_vgGd · 2025-11-01

**Soundness:** 3
**Presentation:** 3
**Contribution:** 3
**Rating:** 6
**Confidence:** 4

**Summary:**

This paper introduces Online Pseudo-Zeroth-Order (OPZO) training, a novel method for supervised learning of Spiking Neural Networks (SNNs). OPZO aims to be more biologically plausible and hardware-friendly than standard backpropagation (BP). It achieves spatial credit assignment using only a single forward pass with injected noise and direct top-down feedback signals via momentum-based feedback connections, avoiding the biologically implausible weight symmetry and separate forward-backward phases of BP. Experiments on various datasets show OPZO achieves competitive accuracy compared to spatial BP while being estimated to have lower computational costs on neuromorphic hardware.

**Strengths:**

1.  The proposed momentum feedback connections successfully address the high variance problem typical of zeroth-order methods, enabling effective training where a basic single-point zeroth-order method fails. This is supported by theoretical analysis and empirical results showing gradient variances comparable to BP.

2.  The method is designed with neuromorphic computing constraints in mind. It requires only one forward pass, enables parallel updates, and aligns with three-factor Hebbian learning rules, potentially leading to lower memory and operation costs for on-chip training compared to BP.

**Weaknesses:**

1.  The biological plausibility claims lack specificity. The paper frequently states that OPZO is more biologically plausible but does not sufficiently elaborate on how specific components, like the momentum feedback connections, map to known biological mechanisms. A more detailed discussion comparing the algorithm's components (e.g., noise injection, top-down signals) to neuroscientific evidence would strengthen this claim.

2.  Theoretical grounding for momentum feedback could be stronger. While Proposition 4.1 analyzes variance reduction, the paper acknowledges that the momentum-based Jacobian estimation can introduce bias. Proposition 4.3 discusses conditions for providing a descent direction, but it would be strengthened by empirical analysis of this bias during training on complex tasks to show it does not hinder convergence.

3.  Integration with local learning is loosely defined. The method of combining OPZO with Local Learning (LL) and Intermediate Global Learning (IGL) is described at a high level. The paper states that for LL, "we assume the weight symmetry for propagating errors," but provides limited justification for this assumption in a biologically plausible or hardware-friendly context. A more detailed mechanism for how these global and local signals are integrated without violating the intended constraints would improve the methodology.

4. The experiment with a 9-layer convolutional network uses Local Learning and IGL. However, the results in Appendix D.5 show that pure OPZO without residual connections performs poorly on deeper networks. This suggests the core method may have scalability limitations, which are mitigated by auxiliary techniques. A clearer discussion of this dependency and more experiments probing the depth limitations of the core OPZO method would provide a more complete picture.

5.  Hardware Cost Analysis is Speculative. The analysis of computational costs (Table 5) is a theoretical estimate for potential neuromorphic hardware. The paper states that "no implementation on neuromorphic hardware is included due to our limited access," which is acknowledged as a limitation. Concrete measurements or more detailed simulations based on specific hardware architectures (e.g., Loihi, SpiNNaker) would make the claims about efficiency much stronger.

**Questions:**

See Weaknesses.

---

> ### Author Response · Authors · 2025-11-20
> **Response to Reviewer vgGd (1/2)**
>
> Thank you for appreciating our work and providing valuable comments. We respond to your comments and questions as follows.
>
> 1. Comparing the algorithm’s components to neuroscientific evidence.
>
> Thank you for your valuable suggestion. We supplement more discussion about noise injection and top-down signals considering neuroscientific evidence.
>
> (1) Noise injection. The biological systems are inherently noisy and it has long been recognized that noise can be utilized as a resource for computation and learning [1,2,3,4]. Considering the perturbation with noise injection, it can be related to stochastic synaptic transmission [1] or “empiric” synapses carrying perturbing input from another part of the brain [2]. Such mechanisms provide the biological basis for perturbation learning, which is believed to be employed by the brain for some kinds of learning [4]. Our work builds on this zeroth-order perturbation, while introducing momentum feedback to solve the large variance problem of it.
>
> (2) Top-down feedback. In the three-factor Hebbian learning, synaptic updates are modulated by reward-prediction errors (RPE) and can be gated by feedback (FB) from higher brain regions through top-down feedback connections, leading to the update rule $\Delta w_{i,j}=\beta\cdot f_i(a_i)\cdot f_j(a_j)\cdot RPE\cdot FB_j$ [5]. Anatomically, feedback projections originate from higher cortical areas and mostly provide input to superficial (L1–L3) and deep (L5) layers of lower sensory areas, targeting apical dendrites of pyramidal neurons and specific microcircuits [5]. These feedback pathways are thought to play a key role in gating plasticity, credit assignment, and context-dependent modulation. The momentum feedback connections in our method are analogous to these top-down feedback pathways that modulate prediction errors, providing a basis for rules similar to three-factor Hebbian learning. Notably, the update of our feedback connections depends only on local pre- and post-synaptic, maintaining the simplicity and biological plausibility of the learning process.
>
> We have added these discussions in Appendix E.3.
>
> 2. Grounding for momentum feedback and empirical analysis of the bias.
>
> Thank you for your suggestion. In our theoretical analysis, we formulate the momentum feedback as dynamically approximating the expectation of the Jacobian of the model function, and provide results on variance reduction and conditions for descent direction. Following your advice, we further supplement the empirical analysis of the bias during training.
>
> Specifically, we supplement the cosine similarity between OPZO and BP (with surrogate gradients) as well as between DFA and BP on CIFAR-100 in Appendix D.3 (Figure 4). As shown in the results, the cosine similarity of different layers between OPZO and BP remains in the range of 0.5-0.9 throughout training, whereas DFA and BP is typically below 0.1 for most layers. This indicates that the bias introduced by momentum feedback does not significantly distort the gradient direction compared to other feedback methods, and the training can converge with effective descent directions (see Appendix D.3, Figure 3).
>
> 3. Detailed discussion of local learning.
>
> Our local learning is based on a linear readout $\mathbf{r}^l=\mathbf{R}^l \mathbf{s}^l$ to calculate the local loss $\mathcal{L}(\mathbf{r}^l, \mathbf{y})$ and the gradient $\frac{\partial \mathcal{L}(\mathbf{r}^l, \mathbf{y})}{\partial \mathbf{s}^l}={\mathbf{R}^l}^{\top}\frac{\partial \mathcal{L}(\mathbf{r}^l, \mathbf{y})}{\partial \mathbf{r}^l}$. The assumption of weight symmetry here refers to using a feedback weight $\mathbf{P}^l={\mathbf{R}^l}$ to propagate errors as ${\mathbf{P}^l}^{\top}\frac{\partial \mathcal{L}(\mathbf{r}^l, \mathbf{y})}{\partial \mathbf{r}^l}$. This is because for the single linear layer that directly connects to the output, the weight $\mathbf{P}^l$ can be learned to be symmetric to $\mathbf{R}^l$ through symmetric local Hebbian-like update rule, i.e., both of them are updated by $\frac{\partial \mathcal{L}(\mathbf{r}^l, \mathbf{y})}{\partial \mathbf{r}^l} {\mathbf{s}^l}^{\top}$ based on pre- and post-synaptic information (e.g., $\frac{\partial \mathcal{L}(\mathbf{r}^l, \mathbf{y})}{\partial \mathbf{r}^l}=\mathbf{r}^l -\mathbf{y}$ for MSE loss and $\frac{\partial \mathcal{L}(\mathbf{r}^l, \mathbf{y})}{\partial \mathbf{r}^l}=\sigma(\mathbf{r}^l) -\mathbf{y}$ for CE loss). This mechanism does not require global error information and is compatible with the intended constraints, while it is only applicable to the single (linear) layer condition.
>
> We have added this explanation in Appendix C.1.

---

> > ### Author Response · Authors · 2025-11-20
> > **Response to Reviewer vgGd (2/2)**
> >
> > 4. The scalability to deeper networks.
> >
> > As discussed in Appendix D.5, pure OPZO has more reliance on the proper network structure (residual connections) than BP for scalability. We have systematically analyzed the scalability of OPZO in deeper networks in Appendix D.5, and our results demonstrate that while pure OPZO without auxiliary mechanisms exhibits degraded performance as network depth increases in plain networks, the introduction of residual connections effectively mitigates this issue and enables scaling to 34 layers. This is consistent with our theoretical insights that the condition for effective descent direction depends on the smoothness of the network, and residual connections can substantially improve the smoothness. We have further revised the main text (Section 5.4) to provide a more complete picture.
> >
> > 5. Hardware cost analysis.
> >
> > Thank you for your question. Our theoretical hardware cost estimates are based on how the feedback pathways for BP, DFA, and OPZO could be mapped onto typical neuromorphic hardware’s architecture such as Loihi 2: (1) BP requires a sequential, layer-by-layer backward pass with symmetric feedback connections. On Loihi 2, this would correspond to constructing dedicated backward pathways for each layer with additional backward synaptic groups, with error signals propagated stepwise from the output layer back to each preceding layer. This sequential feedback routing is not natively supported by Loihi’s event-driven, parallel architecture and would require additional orchestration. (2) DFA uses fixed random feedback pathways from the output layer directly to each hidden layer. On Loihi 2, these can be implemented as separate top-down synaptic groups, allowing error signals to be broadcast in parallel to all hidden layers via graded spikes. This matches Loihi 2’s native support for parallel, event-driven feedback propagation. (3) OPZO adopts the same parallel feedback pathway structure as DFA, but the feedback weights are learned online using local rules. On Loihi 2, each feedback pathway can be updated independently using the chip’s programmable local learning rules, with error signals delivered in parallel to all layers. With the propagated errors, Loihi 2 then supports programmable three-factor update rules for synaptic parameters. This mapping forms the basis of our complexity and cost estimates in the paper, and we believe it is a reasonable and reliable approach given Loihi 2’s architectural features. A full hardware implementation would require much additional engineering efforts with access to specific hardware, which we leave as future work.
> >
> > [1] Seung H S. Learning in spiking neural networks by reinforcement of stochastic synaptic transmission. Neuron, 2003, 40(6): 1063-1073.
> >
> > [2] Fiete I R, Seung H S. Gradient learning in spiking neural networks by dynamic perturbation of conductances. Physical Review Letters, 2006, 97(4): 048104.
> >
> > [3] Maass, W. Noise as a resource for computation and learning in networks of spiking neurons. Proceedings of the IEEE, 2014, 102(5), 860-880.
> >
> > [4] Lillicrap T P, Santoro A, Marris L, et al. Backpropagation and the brain. Nature Reviews Neuroscience, 2020, 21(6): 335-346.
> >
> > [5] Roelfsema P R, Holtmaat A. Control of synaptic plasticity in deep cortical networks. Nature Reviews Neuroscience, 2018, 19(3): 166-180.

---

### Meta-Review · Area_Chair_TDve · 2025-12-31

**Summary:**

The authors propose Online Pseudo-Zeroth-Order (OPZO) training, a method designed to train Spiking Neural Networks (SNNs) using a single forward pass with noise injection and momentum-based feedback. This approach aims to circumvent the biological plausibility and hardware efficiency issues associated with spatial backpropagation (BP) by decoupling the loss function from the model, thereby significantly reducing the gradient variance typically found in zeroth-order methods.

Reviewers generally recognized the novelty of the pseudo-zeroth-order formulation and its effectiveness in reducing variance compared to standard zeroth-order baselines. However, substantial concerns were raised regarding the method's scalability to deeper networks and large-scale datasets (like ImageNet), the theoretical bias introduced by the momentum feedback, and the specificity of the biological plausibility claims. The authors provided a detailed rebuttal including new experiments on ImageNet, gradient similarity analyses, and deeper theoretical discussions, which largely clarified the method's positioning as a hardware-friendly on-chip learning solution rather than a replacement for offline BP.

**Reviewer Concerns:**

### **Addressed Concerns**:

**Scalability and ImageNet Training**: Reviewer Pjav requested results for training from scratch on ImageNet to test the method's limits. The authors added these results in Appendix D.5, showing that while OPZO (19.0%) significantly outperforms DFA (5.6%), it lags behind BP (52.9%) in this offline setting. They successfully contextualized this by emphasizing OPZO's target application is on-chip fine-tuning/adaptation, where it performs competitively.

**Gradient Bias and Validity**: Reviewers vgGd and Pjav questioned the impact of the bias from the momentum feedback and the "pseudo" formulation. The authors addressed this by providing cosine similarity analysis (Appendix D.3), demonstrating that OPZO gradients maintain a high similarity (0.5-0.9) to BP gradients throughout training, unlike DFA which drops below 0.1, proving the descent direction remains effective.

**Biological Plausibility**: Reviewer vgGd criticized the vagueness of biological claims. The authors added detailed connections to neuroscientific evidence in Appendix E.3, specifically mapping noise injection to stochastic synaptic transmission and feedback connections to apical dendrite gating.

**Temporal Gradient Constraints**: Reviewer Pjav asked if the method was fundamentally limited to truncated online settings. The authors provided new results (Full-PZO) showing the method works effectively without truncating temporal gradients, confirming its flexibility.

**Hardware Cost Analysis**: Reviewer vgGd felt the cost analysis was speculative. The authors clarified the mapping to Loihi 2, explaining how the method leverages parallel event-driven feedback which is native to the hardware, unlike the sequential needs of BP.

### **Remaining Concerns**:

**Clean Accuracy Gap**: Reviewer EGrJ and Pjav noted that the method does not outperform BP and struggles with training large networks from scratch. While the authors argue the goal is efficiency and on-chip adaptation, the significant performance gap on ImageNet from scratch (approx. 34% drop vs BP) remains a factual limitation for general-purpose training.

**Dependence on Network Smoothness**: The rebuttal and new experiments confirmed that pure OPZO relies on residual connections to scale to deeper networks because the gradient bias depends on network smoothness. This dependency confirms a limitation in the core estimator's robustness compared to true gradient methods on non-smooth landscapes.

**Reviewer Scores:**

**Reviewer vgGd: 6 $\rightarrow$ 6/7.**

Rationale: The authors provided the specific biological grounding and hardware mapping clarifications requested. The reviewer likely appreciates the detailed response but may maintain the score due to the continued lack of physical hardware implementation.

**Reviewer bmzH: 8 $\rightarrow$ 8.**

Rationale: This reviewer explicitly stated they would stick with their score and did not engage deeply with the technical details.

**Reviewer EGrJ: 4 $\rightarrow$ 4/5.**

Rationale: This reviewer focused on the performance gap with SOTA. While the authors argued their focus is efficiency, the large gap in the new ImageNet experiments reinforces the reviewer's concern about performance, making a significant score increase unlikely.

**Reviewer Pjav: 6 $\rightarrow$ 7.**

Rationale: This reviewer was very constructive, asking for ImageNet scratch results and temporal comparisons. The authors provided exactly the requested data. The reviewer acknowledged the detailed rebuttal, suggesting a positive score adjustment is likely.

---

### Decision · Program_Chairs · 2026-01-26

Accept (Poster)